# NFIC regulates ribosomal biology and ER stress in pancreatic acinar cells and restrains PDAC initiation

Isidoro Cobo[1,2], Sumit Paliwal [1,9], Cristina Bodas [1,2,9], Irene Felipe [1,2,9], Júlia Melià-Alomà[1,3], Ariadna Torres[1,3], Jaime Martínez-Villarreal [1], Marina Malumbres [4], Fernando García [5], Irene Millán[1,2], Natalia del Pozo[1,2], Joo-Cheol Park [6], Ray J. MacDonald[7], Javier Muñoz [5], Raúl Méndez [4,8] & Francisco X. Real [1,2,3] ✉

Pancreatic acinar cells rely on PTF1 and other transcription factors to deploy their transcriptional program. We identify NFIC as a NR5A2 interactor and regulator of acinar differentiation. NFIC binding sites are enriched in NR5A2 ChIP-Sequencing peaks. *Nfic* knockout mice have a smaller, histologically normal, pancreas with reduced acinar gene expression. NFIC binds and regulates the promoters of acinar genes and those involved in RNA/protein metabolism, and *Nfic* knockout pancreata show defective ribosomal RNA maturation. NFIC dampens the endoplasmic reticulum stress program through binding to gene promoters and is required for resolution of Tunicamycin-mediated stress. NFIC is down-regulated during caerulein pancreatitis and is required for recovery after damage. Normal human pancreata with low levels of NFIC transcripts display reduced expression of genes down-regulated in *Nfic* knockout mice. NFIC expression is down-regulated in mouse and human pancreatic ductal adenocarcinoma. Consistently, *Nfic* knockout mice develop a higher number of mutant *Kras*-driven pre-neoplastic lesions.

Pancreatic acinar cells are highly specialized protein synthesis factories that have a well-developed rough endoplasmic reticulum (ER), a prominent Golgi complex, and abundant secretory granules[1]. Acinar differentiation is contingent on the activity of a master regulator, the adult PTF1 complex, composed of the pancreas-specific transcription factors (TFs) PTF1A and RPBJL and the ubiquitous protein E47[2,3]. PTF1 binds the proximal promoter of genes coding for digestive enzymes, secretory proteins and other TFs, and activates their expression. The PTF1 complex is the main driver of acinar differentiation but additional TF with tissue-restricted expression patterns are implicated in the fine-tuning of this process, including GATA6[4], MIST1[5], and NR5A2/LRH-1[6,7]. Acinar cells play a crucial role in acute and chronic pancreatitis, two common and disabling conditions. Genetic mouse models have shown that, upon expression of mutant *Kras*, acinar cells can be the precursors of Pancreatic Intraepithelial Neoplasia (PanIN) and pancreatic ductal adenocarcinoma (PDAC)[8,9].

Our laboratory and others have shown that the acinar differentiation program acts as a tumor suppressor in the pancreas. Monoallelic or homozygous inactivation of several acinar transcriptional regulators in the germline, the embryonic pancreas, or the adult

[1]Epithelial Carcinogenesis Group, Spanish National Cancer Research Centre-CNIO, Madrid, Spain. [2]CIBERONC, Madrid, Spain. [3]Departament de Medicina i Ciències de la Vida, Universitat Pompeu Fabra, Barcelona, Spain. [4]Institute for Research in Biomedicine (IRB Barcelona), The Barcelona Institute of Science and Technology, Barcelona, Spain. [5]Proteomics Unit, Spanish National Cancer Research Centre-CNIO, ProteoRed-Instituto de Salud Carlos III, Madrid, Spain. [6]Department of Oral Histology-Developmental Biology, School of Dentistry, Seoul National University, Seoul, Korea. [7]Department of Molecular Biology, University of Texas Southwestern Medical Center, Dallas, TX, USA. [8]Institució Catalana de Recerca i Estudis Avançats (ICREA), Barcelona, Spain. [9]These authors contributed equally: Sumit Paliwal, Cristina Bodas, Irene Felipe. ✉e-mail: preal@cnio.es

pancreas can result in compromised acinar function that favors loss of cellular identity and poises acinar cells for transformation upon activation of mutant *Kras*[10–12]. The tumor suppressive function of these TF is not obvious because the exocrine pancreas has a large functional reserve, i.e. massive alterations in cellular function need to occur in order to be reflected in histological or clinical changes. Furthermore, the context in which loss of function takes place has an important impact on the phenotype[13].

NFIC is a member of the nuclear factor I family of TFs that regulate both ubiquitous and tissue-restricted genes[14]. In the mammary gland, NFIC activates the expression of milk genes involved in lactation[15]. Furthermore, it acts as a breast cancer tumor suppressor, as it directly represses the expression of *Ccnd1* and *Foxf1*, a potent inducer of epithelial-mesenchymal transition (EMT), invasiveness, and tumorigenicity[16,17]. Additional roles have been proposed through the regulation of *Trp53*[18]. The physiological role of NFIC has been best studied in dentinogenesis, since *Nfic*[-/-] mice develop short molar roots and display aberrant odontoblast differentiation and dentin formation[19]. NFIC regulates odontoblast-related genes, including *Dssp*[20], Wnt[21], and hedgehog signaling[22]. These defects result in growth retardation and reduced survival upon standard feeding. NFIC also regulates postnatal growth and regeneration in other tissues[23,24] and its deficiency results in delayed growth and reduced survival[19].

Here, using a combination of omics analyses and studies in knockout mice and cultured cells, we uncover the role of NFIC as a regulator of acinar function whose major impact is at the level of the ER stress response in murine and human pancreas. Unlike most other TFs previously identified as required for full acinar function, NFIC belongs to a family of acinar regulators with tissue-wide expression. NFIC dysregulation sensitizes the pancreas to damage and neoplastic transformation.

## Results

### Identification of transcription factors involved in the regulation of pancreatic acinar differentiation

To discover additional transcription factors that might cooperate with known acinar regulators (e.g. PTF1A, GATA6, NR5A2, and MIST1), we reanalyzed publicly available ChIP-sequencing data and used HOMER to search for motifs enriched in the sequencing reads. As expected, the cognate binding sites of these factors were the top enriched motif in each respective analysis (Fig. 1A). Motifs corresponding to RBPJ/RBPJL and HNF1, known regulators of acinar differentiation, were also enriched, thus validating the strategy applied. In addition, we found consistent enrichment of the NF1(CTF)/NFIC motif across all the experiments, with lowest p-values in the NR5A2 ChIP-Seq dataset. This motif is significantly enriched in NR5A2 ChIP-Seq peaks from normal adult mouse pancreas[25] (Fig. 1A) but not from mouse ES cells[25], pointing to lineage identity specificity. Analysis of the spacing between NR5A2 and NFIC motifs in the genomic regions bound by NR5A2, using the SpaMo tool from MEME suite, showed that NFIC and NR5A2 binding motifs are located close to each other, with a spacing of 29 nucleotides being the most significantly conserved distance ($P =$ e-16) (Fig. 1B). To assess whether NR5A2 interacts with NFI family members, we immunoprecipitated normal mouse pancreas lysates with anti-NR5A2 antibodies, followed by mass-spectrometry analysis, and identified NFIC as one of the top significantly enriched proteins (Fig. 1C and Supplementary Data 1). Using immunoprecipitation and western blotting, the interaction was confirmed in normal pancreas from adult —but not E17.5—mice (Fig. 1D); as expected, NR5A2 immunoprecipitates from adult *Nfic*[-/-] pancreata did not contain NFIC (Supplementary Fig. 1A). The major role of NFIC, rather than other related family members, is also supported by the fact that NFIC is expressed at highest levels both in mouse and human pancreas (Fig. 1E). Analysis of published ChIP-Seq data revealed several NR5A2 and PTF1A peaks in the proximal *Nfic* promoter and the binding was confirmed by ChIP-

qPCR (Fig. 1F), strongly suggesting that *Nfic* is a PTF1A and NR5A2 target. Using ChIP-qPCR, NFIC was found to bind the promoter of bona fide acinar genes such as *Cela2a*, *Cpa1*, *Ctrb1*, *Pnlip*, and *NrOb2* that were similarly bound by NR5A2 (Fig. 1G).

To assess the cellular distribution of NFIC, we performed immunohistochemistry (IHC) with a well-validated antibody that lacks reactivity with tissues of *Nfic*[-/-] mice (Supplementary Fig. 1B). In normal 8-week-old pancreas, NFIC is expressed at high levels in acinar cells and at lower levels in endocrine and ductal cells (Fig. 1H). These results were confirmed using triple immunofluorescence (IF) with antibodies detecting PTF1A, INS1, and KRT19 (Supplementary Fig. 2). NFIC was undetectable at E12.5 and E14.5 in CDH1;[+]PTF1A[+] pancreatic progenitors (Supplementary Fig. 3A, B) but it was detected at E16.5 and E18.5 in PTF1A[+] acinar as well as in KRT19- and INS1-expressing cells (Supplementary Fig. 3C, D).

To determine whether NFIC is required for the expression of digestive enzyme transcripts, we knocked down *Nfic* in 266-6 acinar cells using lentiviral shRNAs: a significant down-regulation of *Cela2a* and *Ctrb1*—as well as *Ptf1a* and *Rbpjl*—was demonstrated. Expression of PTF1A, CTRB1, and CPA proteins was similarly reduced (Fig. 1I). In addition, NFIC expression in HEK293 cells resulted in increased activity of an *Ela1b* promoter-reporter luciferase construct (Fig. 1J). Altogether, the data support an important role for NFIC in the regulation of the acinar program after the secondary transition.

### NFIC is part of the transcriptional network responsible for acinar identity and function

To further assess the role of *Nfic* in pancreatic development and homeostasis, we used constitutive *Nfic* knockout (*Nfic*[-/-]) mice[19]. Because of their dental phenotype, their maintenance requires special handling that was similarly applied to all mice, regardless of the genotype, as described in the Methods section. At age 8 weeks, *Nfic*[-/-] mice are viable and have a normal weight but their pancreas is significantly smaller (pancreas/body weight ratio) than that of WT mice (Fig. 2A). We did not observe significant differences in the proportion of exocrine pancreas across genotypes (WT = 96.98 ± 1.61, KO = 94.46 ± 1.67; *P*-val = 0.18), nor in the expression of INS1, KRT19, and SOX9 (Supplementary Fig. 4A). Results of glucose tolerance tests were similar in 8-week-old control and *Nfic*[-/-] mice, except that glucose levels were reduced by 60–120 min in the latter (Supplementary Fig. 4B,C). Histological and IHC analysis of the pancreas at 8 weeks showed a significant increase of CD45[+] and Ki67[+] cells in the pancreas of *Nfic*[-/-] mice, but the proportion of cells undergoing apoptosis (detected as cells expressing cleaved caspase 3) was not significantly different (Supplementary Fig. 5A). Interestingly, the pancreas of 20–25-week-old mice was histologically normal and there were no significant differences in the percentage of CD45[+], Ki67[+], or cleaved caspase 3[+] cells (Supplementary Fig. 5B).

Because histology lacks sensitivity to disclose subtle alterations in exocrine function[4,10,26] we performed RNA-Seq of pancreata from 8–10-week-old wild type and *Nfic*[-/-] mice (Fig. 2B). We identified 1641 and 1568 transcripts that were significantly up- and down-regulated, respectively, in *Nfic*[-/-] pancreata (Supplementary Data 2). Multiple genes belonging to the exocrine differentiation program were among the down-regulated transcripts (e.g. *Ptf1a*, several digestive enzymes), as validated using RT-qPCR (Fig. 2C and Supplementary Fig. 6A). The activity of the acinar signature reported by Masui et al.[27] was also significantly down-regulated (Supplementary Fig. 6B). Exemplary down-regulated genes include *Amy2a5*, *Bhlha15*, *Cel*, *Cela1*, *Cela2b*, *Cpa1*, *NrOb2*, and *Pnliprp1*. Reduced levels of corresponding proteins were demonstrated using western blotting and immunofluorescence (Fig. 2D, E and Supplementary Fig. 6C). Selected changes were validated by RT-qPCR in freshly isolated acini (Fig. 2F). Reduced expression of digestive enzyme transcripts was also detected in E17.5 KO pancreata, indicating that NFIC loss impacts on acinar differentiation prenatally (Supplementary Fig. 6D). The RNA-Seq data also showed

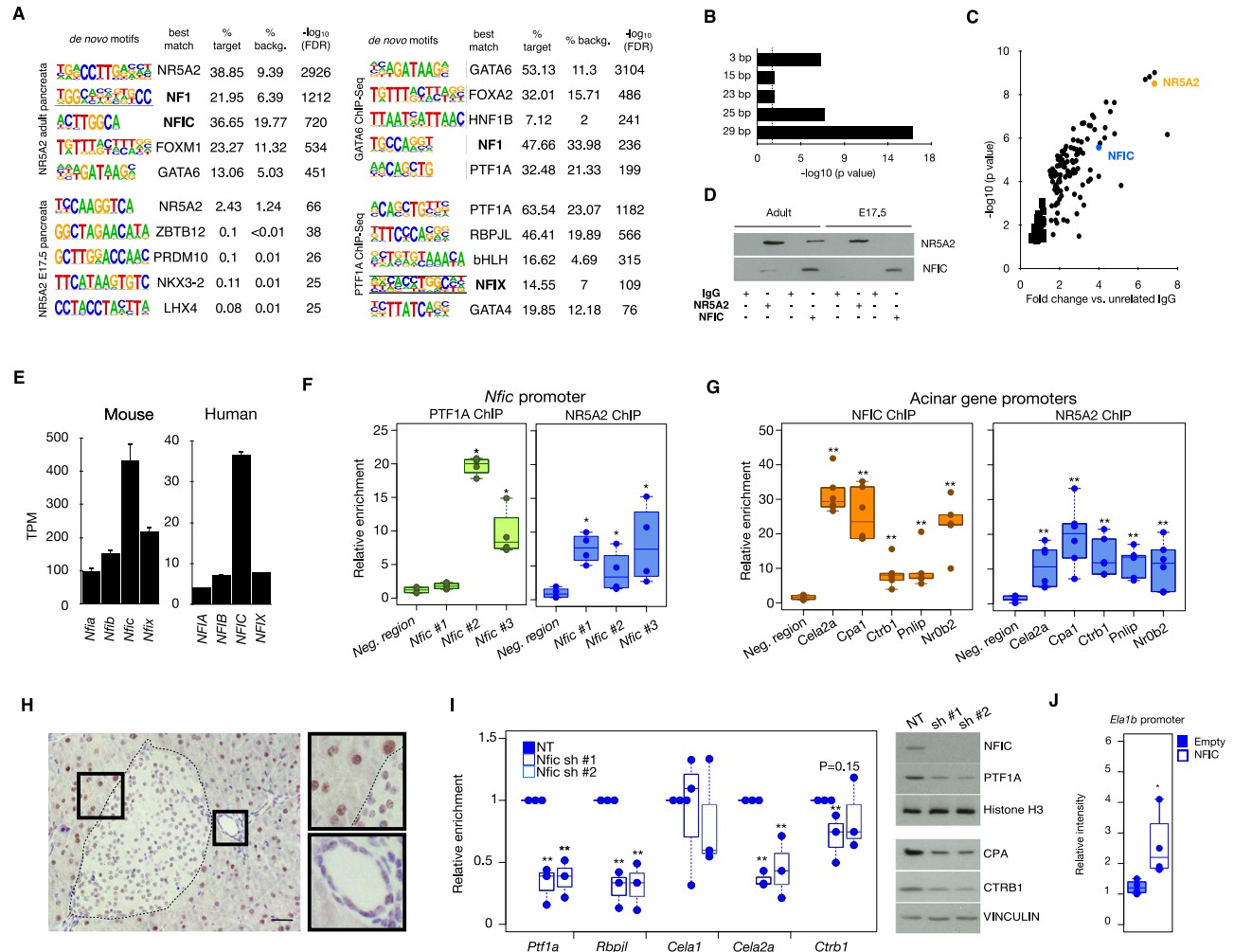

**Fig. 1 | NFIC is an NR5A2 interactor and an acinar regulator. A** HOMER de novo motif analysis for NR5A2, PTF1A and GATA6 ChIP-Seq in mouse pancreata showing enrichment in NF1/NFI motifs. **B** SpaMo analysis showing distance conservation of the NR5A2 and NFIC motifs in the regions bound by NR5A2. **C** IP-MS analysis using lysates from normal mouse pancreata reveals that NFIC is among the top NR5A2 interactors identified (one-tailed T-test with a permutation-based FDR control was used for statistical analysis). **D** Immunoprecipitation-western blotting analysis showing that NFIC and NR5A2 are part of the same complex in adult, but not in embryonic, pancreas (one representative image of 2 independent experiments shown). **E** Expression of *NFI* transcripts in mouse (left panel, *n* = 4 replicates) or human (right panel, GTEX *n* = 328 samples) pancreata assessed by RNA-Seq showing that *NFIC* is the family member expressed at highest levels. **F** ChIP-qPCR shows binding of NR5A2 and PTF1A at the *Nfic* promoter (one region in NR5A2 peak1 and two regions in NR5A2 peak3), compared to a control (Neg) region (normalized to unrelated IgG) (*n* = 4/group, two-tailed Mann–Whitney *U*-test) (Nfic #1: PTF1A, *P* = 0.3429; NR5A2, *P* = 0.028) (Nfic #2: PTF1A, *P* = 0.028; NR5A2, *P* = 0.048) (Nfic #3: PTF1A, *P* = 0.028; NR5A2, *P* = 0.028). **G** ChIP-qPCR of NR5A2 and NFIC binding to the promoter of digestive enzyme genes and *NrOb2*; controls as in panel F (*n* = 6/

group, two-tailed Mann-Whitney *U*-test) (NR5A2 ChIP: *Cela*, *Cpa1*, *Ctrb1*, *Pnlip* and *NrOb2*, *P* = 0.002; NFIC ChIP: *Cela*, *Ctrb1*, *Pnlip* and *NrOb2*, *P* = 0.002; *Cpa1*, *P* = 0.005). **H** IHC analysis of NFIC expression in normal adult mouse pancreas showing higher expression in acinar cells and lower expression in endocrine and ductal cells (insets) (One representative image of 4 replicates is shown, scale bar = 10 μm). **I** Lentiviral *Nfic* knockdown in 266-6 cells showing reduced expression of transcripts coding for digestive enzyme transcripts and pancreatic TFs (RT-qPCR) (left panel, two-tailed Mann–Whitney *U*-test); western blotting analysis of the corresponding samples interfered with non-targeting (NT) or *Nfic*-targeting shRNAs (*n* = 3/group; one representative image of 3 independent experiments is shown). (*Ptf1a* in *Nfic* sh#1, *P* = 0.003 and in *Nfic* sh#2, *P* = 0.002; *Rbpjl* in *Nfic* sh#1, *P* = 0.0001 and in *Nfic* sh#2, *P* = 0.0004; *Cela2a* in *Nfic* sh#1, *P* = 0.0012 and in *Nfic* sh#2, *P* = 0.009; *Ctrb1* in *Nfic* sh#1, *P* = 0.01 and in *Nfic* sh#2, *P* = 0.15). **J** *Ela1b*-luciferase promoter-reporter analysis shows increased activity upon expression of NFIC in HEK293 cells (*n* = 4/group) (*P* = 0.029). Barplots are presented as mean values +/− SD. Boxplots show the minimum, the maximum, the sample median, and the first and third quartiles. Source data are provided as a Source Data file.

reduced expression of genes involved in epithelial polarity (e.g., *Muc1*) and cell adhesion (e.g. *Cdh1*) and up-regulation of transcripts coding for EMT markers (e.g., *Vimentin*, *Twist*, and *N-cadherin/Cdh2*) (Supplementary Data 2). The down-regulation of CDH1 was confirmed using immunohistochemistry (Supplementary Fig. 5B) and is in accordance with findings in dentinogenesis[20]. NR5A2 expression was similar in control and *Nfic*[-/-] pancreata (Fig. 2D), indicating that the effects of *Nfic* inactivation are not secondary to changes in NR5A2 expression.

We quantified the overlap of differentially expressed genes in *Nfic*[-/-] pancreata with that in mice in which *Nr5a2*[6], *Ptf1a*[28] or *Mist1*[26,29–34] has been inactivated in the pancreas (PKO). We found a significant

overlap of genes down-regulated in the pancreas of *Nfic*[-/-] and *Nr5a2* pancreas-knockout, *Ptf1a* pancreas- knockout, or *Mist1* knockout mice [41% (52/126), 46.97% (231/492), 36.33% (262/721), respectively] but not of the up-regulated genes [2.36% (3/127), 9.99% (41/414), 16.37% (75/458), respectively] (Fig. 2G) These findings suggest that NFIC is a member of the acinar transcription factor network.

## Integration of ChIP-Seq and RNA-Seq data reveals broader functions of NFIC in the pancreas

To unveil the transcriptional program driven by NFIC, and understand the mechanisms involved, we performed ChIP-seq using pancreata

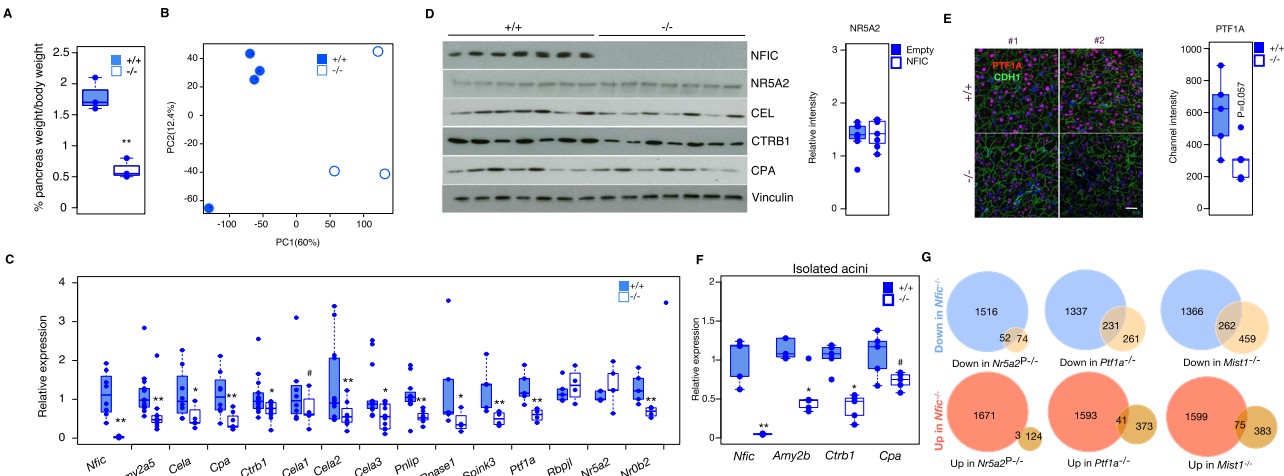

**Fig. 2 | NFIC is required for normal acinar cell differentiation. A** The pancreas of *Nfic*[-/-] mice has a reduced relative size (*n* = 3/group, two-tailed Student *T*-Test, *P* = 0.0013). **B** Principal component analysis of the RNA-Seq transcriptomes from WT and *Nfic*[-/-] mice. **C** RT-qPCR showing reduced expression of transcripts coding for digestive enzymes and pancreatic TF in *Nfic*[-/-] pancreata [*n* = 3/group; *P* < 0.1 (#), *P* < 0.05 (*), *P* < 0.01 (**), two-tailed Mann Whitney *U*-Test. *Nfic*, *P* = 0.0015; *Amy2a5*, *P* = 0.0006; *Cela*, *P* = 0.011; *Cpa*, *P* = 0.009; *Ctrb1*, *P* = 0.034; *Cela1*, *P* = 0.2187; *Cela2*, *P* = 0.024; *Cela3*, *P* = 0.028; *Pnlip*, *P* = 0.003; *Rnase1*, *P* = 0.25; *Spink3*, *P* = 0.035; *Ptf1a*, *P* = 0.015; *Rbpjl*, *P* = 0.73; *NrSa2*, *P* = 0.015; *NrOb2*, *P* = 0.035]. **D** Western blotting showing reduced expression of digestive enzymes, but not NR5A2 (*P* = 0.654), in *Nfic*[-/-] pancreata (*n* = 7/group). **E** IF analysis of the expression of PTF1A in pancreatic epithelial CDH1+ cells of *Nfic* WT and KO mice. (Scale bar = 10 μm). Fluorescence quantification shown in the accompanying bar graph (*P* = 0.03, two-tailed Student *T*-test). Densitometric quantification of panel 3B: band intensity normalized to loading control, relative to wild-type pancreata (*n* = 5/group). **F** RT-qPCR showing reduced expression of transcripts coding for digestive enzymes in primary acini from *Nfic*[-/-] mice [*n* = 5/group, *P* < 0.1 (#), *P* < 0.05 (*), *P* < 0.01 (**); two-tailed Mann−Whitney *U*-test. *Nfic*, *P* = 0.008; *Amy2b*, *P* = 0.015; *Ctrb1*, *P* = 0.007; *Cpa*, *P* = 0.095]. **G** Comparison of the overlap of DEG in the pancreas of *Nfic*[-/-] vs. that of mice lacking NR5A2, PTF1A, and MIST1 (details in text). Statistics: two-tailed Student *T*-test. Significant overlap is shown for downregulated genes compared to a random list of genes. "N-1" chi-squared test was used to calculate statistical significance. Barplots are presented as mean values +/− SD. Boxplots show the minimum, the maximum, the sample median, and the first and third quartiles. Source data are provided as a Source Data file.

from 8–10-week-old wild type mice. A total of 15824 peaks bound by NFIC were identified, corresponding to 9086 genes, with enrichment of motifs corresponding to NF1, bHLH proteins, and TF involved in acinar cell differentiation such as FOXA1, GATA6, and nuclear receptors, among others (Fig. 3A). NFIC peaks were enriched in the vicinity of the TSS of genes [−5000; TSS, 20.05%] and the corresponding sequences displayed significant enrichment of NF1, SP2, THAP, ELK1, and AP-1 motifs (Fig. 3B).

RNA-seq and ChIP-seq data were integrated to identify genes/pathways directly regulated by NFIC: 55.54% (871/1568) of the down-regulated genes (*P* < 0.05) and 36.3% (593/1634) of the up-regulated genes in *Nfic*[-/-] pancreata were bound by NFIC (*P* < 0.05). A greater percentage of down-regulated genes (47.19%, 411/871) relative to up-regulated genes (36.42%, 216/593), had NFIC peaks in the promoter region [−1000:TSS] (Fig. 3C). The proportion of NFIC high affinity peaks−as defined by the top two quartiles of peak score (Q1 + Q2)−was also higher in down-regulated genes relative to up-regulated genes (60% vs. 51%, respectively. *P* < 0.05) (Fig. 3D). Gene set enrichment analysis showed that NFIC-bound down-regulated genes were associated with protein metabolism (e.g., oxidative phosphorylation, protein export, ribosome, seleno amino acid, and purine metabolism, among others) (Fig. 3E). The motifs enriched in down-regulated genes with NFIC peaks [−1000;TSS] include NFI, FOXM1, bHLH, RBPJ, and ARID5A (Fig. 3F).

Among the gene sets whose activity was up-regulated in *Nfic*[-/-] pancreata, we found several involved in inflammation/immune response, including chemokines (e.g., *S100a9*, *Ccl5*, *Ccl7*, *Cxcl12*, *Cxcl3*) and complement components (e.g., *C1qb*, *C3*, *Cfb*, *Cfd*) (Supplementary Fig. 7A−C). NFIC-bound up-regulated genes were also enriched in inflammatory pathways (e.g., chemokine signaling, leukocyte transendothelial migration) (Fig. 3G). Accordingly, *Ccl5* and *Cxcl13* transcripts were up-regulated upon *Nfic* knockdown in 266-6 cells (Supplementary Fig. 7D). These results indicate that NFIC contributes to restrain an inflammatory program in the pancreas.

Other relevant gene sets whose activity is up-regulated are those related to adhesion/motility (e.g., focal adhesion, axon guidance, ECM receptor interaction), and signaling (e.g. MAPK, insulin signaling) (Fig. 3G). Motifs enriched in up-regulated genes included NFI, TLX/nuclear receptors, HNF1, TCF3, and CTCF (Fig. 3H). Representative examples of ChIP-Seq findings are displayed in Fig. 3I.

## NFIC distinctly regulates the ribosomal program in the mouse and human pancreas

A striking finding from the enrichment analyses referred to gene sets associated with protein synthesis biology (Fig. 3E and Supplementary Fig. 8A). Multiple transcripts coding for ribosomal proteins were down-regulated in *Nfic*[-/-] pancreata (Supplementary Fig. 8B) and reactivity with an antibody detecting the 5.8 S rRNA−a surrogate readout of ribosomes−was reduced in *Nfic*[-/-] acinar cells (Fig. 4A) but not in islet cells. Several genes coding for ribosomal proteins were found to be bound by NFIC in Chip-Seq experiments with both normal pancreas tissue and in a variety of experiments from the ENCODE project using cultured cells (Supplementary Fig. 8C). We analyzed rRNA synthesis and maturation using RT-qPCR and found a significantly reduced expression of 45 S pre-rRNA, mature 18 S rRNA, 5.8 S rRNA, and 28 S rRNA in the pancreas of *Nfic*[-/-] mice (Fig. 4B), indicating reduced transcription of rRNA genes. There were no significant differences in the immunostaining of the ER marker calreticulin in acinar cells of WT vs. *Nfic*[-/-] mice (mean fluorescence staining: 0.40 ± 0.11 vs. 0.26 ± 0.14; *P* = 0.34). Analysis of protein synthesis in acinar cells from wild type and *Nfic*[-/-] mice using flow cytometry showed a trend, albeit non-significant (*P* = 0.13), towards reduced translation in KO pancreata (Supplementary Fig. 8D). In addition, there was a down-regulation of mitochondrial, ER, and Golgi complex pathways, including altered expression of *Fkbp2, Dio*, and *Pink1* that were bound by NFIC at their promoter region (Supplementary Fig. 8E−G).

To determine whether similar changes occur in normal human pancreas in association with low NFIC, we used the GTEX dataset

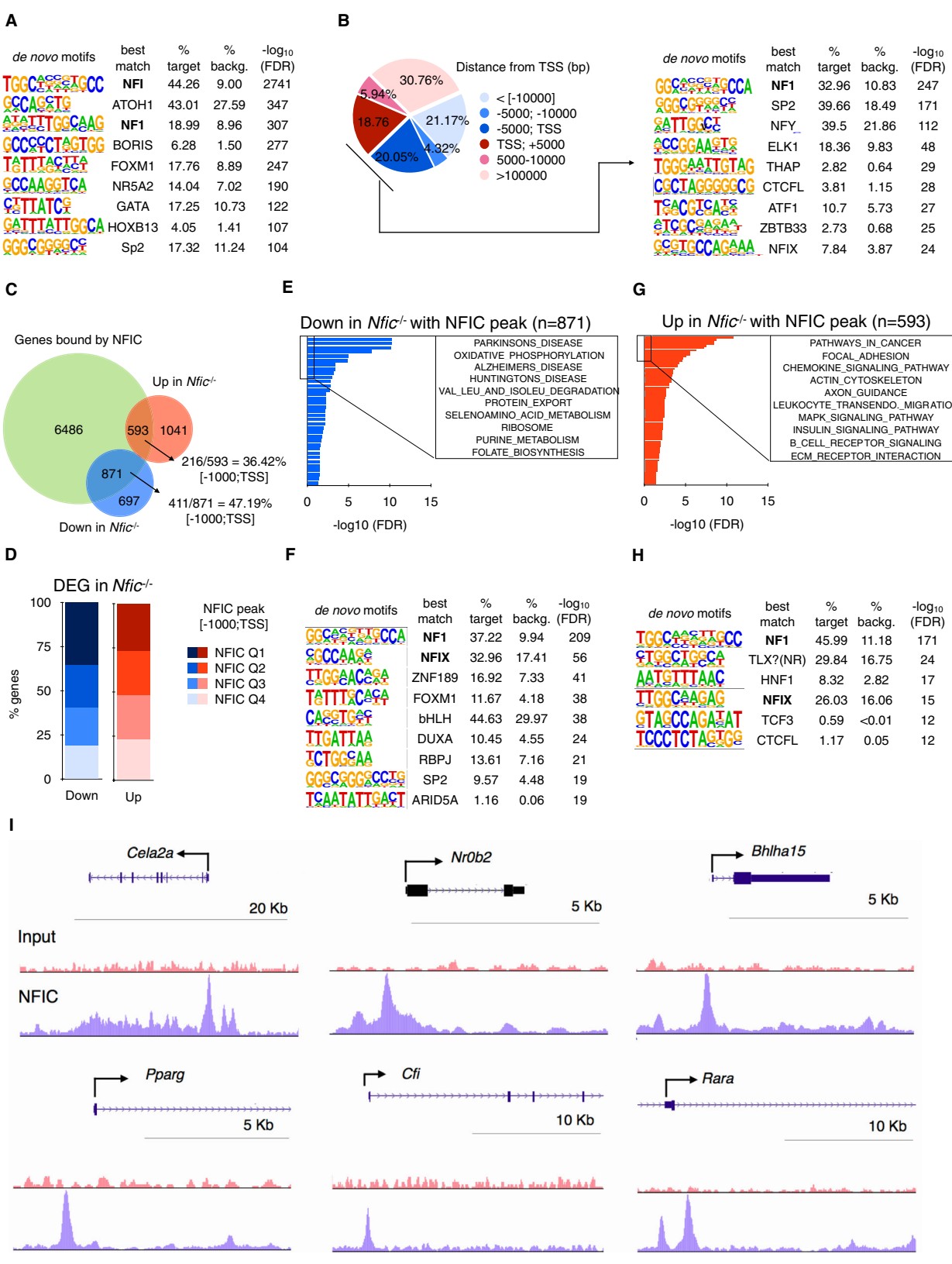

(n = 171) and compared gene expression in samples with high vs. low *NFIC* levels (top vs. bottom 10 individuals): there was a 2.06-fold log2 difference in *NFIC* transcript levels in *NFIC*$^{high}$ vs. *NFIC*$^{low}$ pancreata. Ninety-four percent of genes that were down-regulated in *Nfic*$^{-/-}$ pancreata were also down-regulated in *NFIC*$^{low}$ human pancreas (P = 1.79 e$^{-47}$) compared to 63% of a random gene list (Fig. 4C). Among the common down-regulated genes/pathways are several

involved in ribosomal function, including *RPS5, RPS8, RPS11, RPS15, RPS21, RPS26,* and *RPS29* (Fig. 4D, E). By contrast, only 27% of transcripts with upregulated expression in *Nfic*$^{-/-}$ pancreata were upregulated in *NFIC*$^{low}$ human samples, compared to 37% of a random list of genes (P = 3.47e$^{-8}$). These data support that NFIC loss impacts on ribosomal biology and that this role is conserved in normal human pancreas.

**Fig. 3 | NFIC binds to genomic regions associated to genes involved in acinar differentiation, ER stress, UPR, and inflammation. A** De novo motif analysis of NFIC ChIP-Seq peaks showing NFI as the top-motif; bHLH, NR5A2 and GATA are among the additional top motifs. **B** Distribution of NFIC ChIP-Seq peaks showing binding to regions close to the TSS (left) and the corresponding enrichment of the NFI, ELK and CTCF motifs (right). **C** Venn diagram showing the overlap between genes with an NFIC peak and those de-regulated in the *Nfic*$^{-/-}$ pancreas showing a greater overlap for the down-regulated genes. **D** Bar graph of the distribution of NFIC ChIP-Seq peaks based on score intensity and the overlap with genes de-

regulated in *Nfic*$^{-/-}$ pancreata showing slight greater overlap in Q1, Q2 for the down-regulated genes. **E**–**H** Gene set enrichment analysis of genes bound by NFIC and down-regulated (**E**) or up-regulated (**F**) in *Nfic*$^{-/-}$ pancreata showing downregulation of bona fide acinar, ribosomal, and metabolic genes and up-regulation of pathways related to inflammation and signaling. Motif analysis of genes with an NFIC peak that are down-regulated (**G**) or up-regulated (**H**) in *Nfic*$^{-/-}$ pancreata: NFI is the top motif in both groups. **I** UCSC browser shots of NFIC ChIP-Seq data showing enrichment at the *Cela2a*, *NrOb2*, *Bhlha15*, *Pparg*, *Cfi*, and *Rara* loci. Source data are provided as a Source Data file.

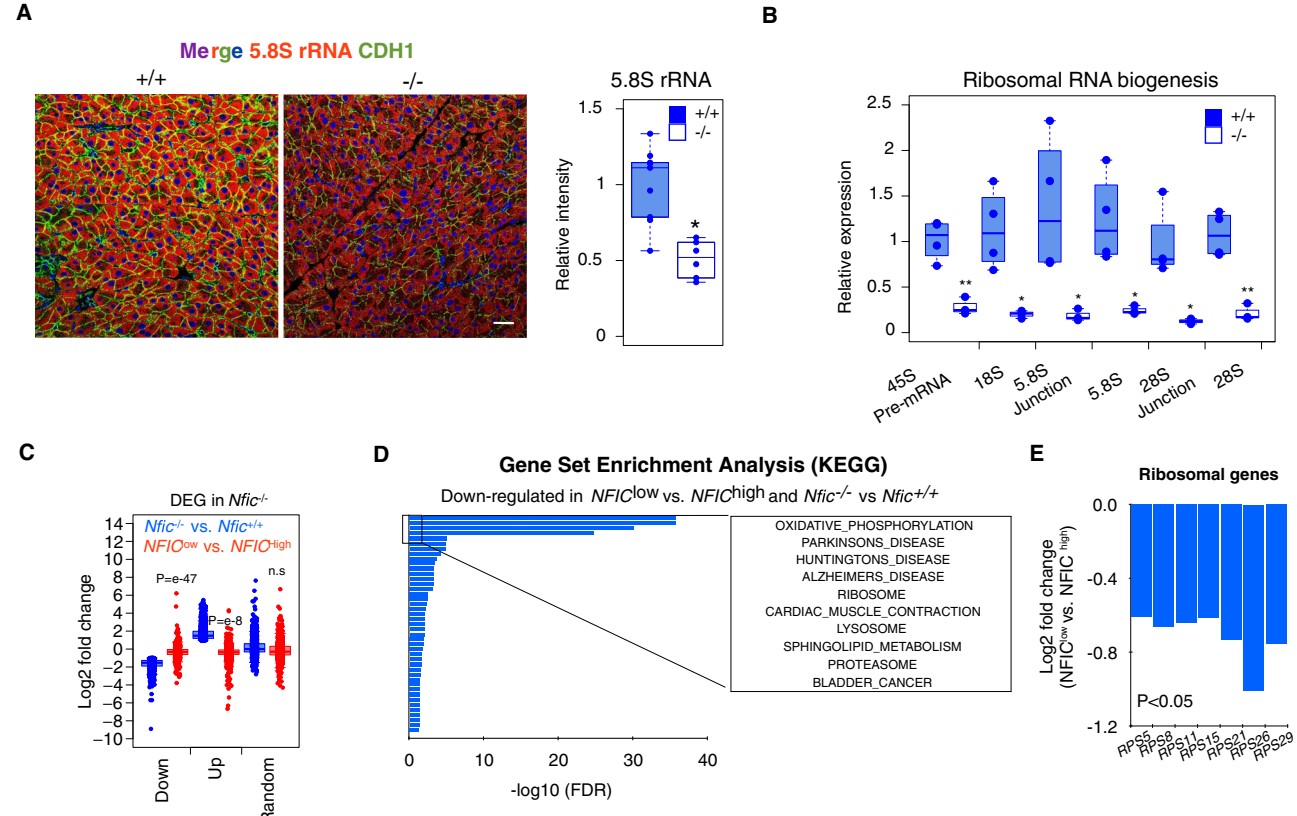

**Fig. 4 | NFIC regulates aspects of protein biosynthesis in the pancreas. A** IF analysis with an antibody recognizing 5.8 S rRNA and CDH1 shows decreased expression of both in *Nfic*$^{-/-}$ acinar cells. (Scale bar = 5 μm). Quantification is shown in the accompanying panel ($n \geq 6$/group, $P < 0.05$ (*), two-tailed Mann–Whitney *U*-test) ($P = 0.038$). **B** RT-qPCR analysis showing altered ribosomal RNA maturation in *Nfic*$^{-/-}$ pancreata ($n = 4$/group; $P < 0.01$ (**), two-tailed Student *T*-test) (*18 S*, $P = 0.01$; *5.8 S Junction*, $P = 0.04$; *5.8 S*, $P = 0.01$; *28 S Junction*, $P = 0.01$; *28 S*, $P = 0.002$; *45 S*, $P = 0.003$. **C** Boxplot plot displaying the relationship between the expression of upregulated, down-regulated, or a random set of genes, in control *Nfic*$^{+/+}$ vs. *Nfic*$^{-/-}$ mice and in histologically normal human pancreatic tissues samples [top 10 low- vs. top 10 high- *NFIC* expressing samples, as determined by RNA-Seq analysis (*NFIC*$^{low}$ vs. *NFIC*$^{high}$)]. Data shows the concordant pattern between down-regulated genes in

*Nfic*$^{-/-}$ mice and *NFIC*$^{low}$ human pancreata. "N-1" two-tailed chi-squared test was used to calculate statistical significance. The *P*-value was calculated comparing to a random gene list. **D** GSEA for genes that are concurrently down-regulated in *Nfic*$^{-/-}$ vs. WT pancreata and in *NFIC*$^{low}$ vs. *NFIC*$^{high}$ human pancreata. Genes were computed with KEGG data sets showing the similarities with those gene sets under-represented in *Nfic*$^{-/-}$ mice. **E** Bar plot displaying the lower expression of genes coding for ribosomal proteins in *NFIC*$^{low}$ vs. *NFIC*$^{high}$ human pancreata (*RPS5*, $P = 3.1$ e-4; *RPS8*, $P = 5.2$ e-5; *RPS11*, $P = 4.1$ e-4; *RPS15*, $P = 7.3$ e-4; *RPS21*, $P = 4.6$ e-7; *RPS26*, $P = 8.2$ e-3; *RPS29*, $P = 5.3$ e-4; an integration of Fisher's exact test and likelihood ratio were used to calculate statistical significance). Barplots are presented as mean values +/– SD. Boxplots show the minimum, the maximum, the sample median, and the first and third quartiles. Source data are provided as a Source Data file.

## NFIC regulates the unfolded protein and ER stress responses

A large number of NFIC-bound and regulated genes are involved in autophagy (e.g. *Ulk1, Prkaa2, Pik3c3, Gabarap, Gabarapl1, Map1lc3b, Sqstm1, Pink1, Dap*), amino acid metabolism, ER stress, and the UPR (e.g. *Dnajc5, Dnajc13, Hsp90aa1*) (Fig. 5A, Supplementary Fig. 9A), pathways that have been shown to be critically relevant in pancreatic homeostasis and disease[26,29–34]. Accordingly, p62/SQSTM1 was over-expressed in a subset of acinar cells from *Nfic*$^{-/-}$ mice (Supplementary Fig. 9B). Alterations in autophagy and the unfolded protein response (UPR) induce ER stress[35]. Up-regulation of *Chop/Ddit3, Hspa5/BiP*, and spliced *Xbp1* (*sXbp1*) transcripts in total pancreas and in isolated acini

of KO mice was confirmed, using RT-qPCR (Fig. 5B, C). We also observed a ca. 2-fold, significant, up-regulation of BIP and CHOP in *Nfic*$^{-/-}$ pancreata; BIP up-regulation in acinar cells was confirmed by IF (Fig. 5D, E). Analysis of the ChIP-Seq data shows that NFIC binds to the proximal promoter and distal region of 28.31% (43/81) and 24.69% (20/81), respectively, of the genes associated to ER stress[36] (e.g. *Ddit4* and *Slc1a5*) (Supplementary Fig. 9C). ChIP-qPCR showed that NFIC—but not NR5A2—bound to the promoter of *Hspa5/BiP, Ddit3/Chop*, and *Hsp90aa1* (Fig. 5F), highlighting that NFIC selectively regulates an aspect of the acinar secretory program related to the ER stress response. We extended these analyses to the normal human GTEX

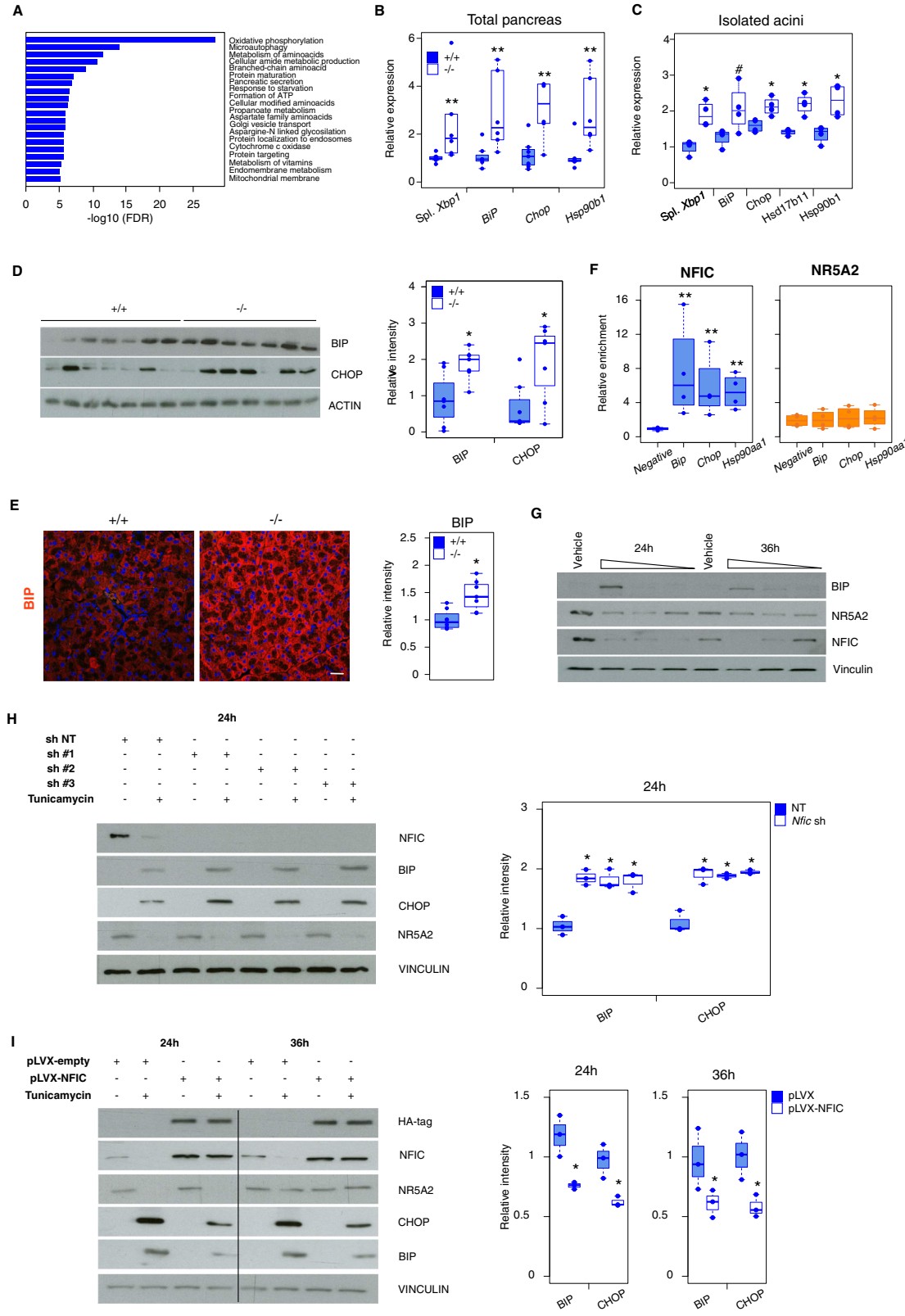

pancreata dataset and found an up-regulation of several UPR genes, including *HSPA90AA1*, *CALR3*, and *HSPA6* among others, in the pancreas of individuals with low levels of *NFIC* (Supplementary Fig. 9D).

To determine whether NFIC is involved in the ER stress response, 266-6 cells were treated with tunicamycin (TM), a protein N-glycosylation inhibitor. As expected, at 24 h we observed a dose-dependent up-regulation of BIP, and a down-regulation of NFIC and

NR5A2; the latter has been shown to participate in the ER stress response in the liver[36]. By 36 h, NFIC levels remained low whereas NR5A2 expression had recovered (Fig. 5G). *Nfic* knockdown in 266-6 cells did not affect basal BIP or CHOP expression but it sensitized cells to the effects of TM (Fig. 5H). Accordingly, NFIC-overexpressing cells showed reduced expression of BIP and CHOP upon treatment with TM (Fig. 5I). In both cases, NR5A2 expression was unaffected, supporting

**Fig. 5 | NFIC regulates UPR and ER stress resolution. A** GSEA analysis of the UPR and ER stress gene sets down-regulated in *Nfic*[-/-] pancreata and up-regulation of UPR. **B, C** RT-qPCR analysis of ER stress-related transcripts in WT and *Nfic*[-/-] pancreata (*n* ≥ 6/group) (**B**) and in freshly isolated acini (*n* ≥ 3/group) (**C**). Two-tailed Mann–Whitney *U*-test was used to calculate statistical significance in **B** and **C**, except for *BiP* in **C** where two-tailed Student *T*-test was used (**B** *Xbp1* spliced, *P* = 0.0038; *BiP*, *P* = 0.009; *Chop*, *P* = 0.008; *Hsp90b1*, *P* = 0.0101) (**C** *Xbp1* spliced, *P* = 0.028; *BiP*, *P* = 0.005; *Chop*, *P* = 0.008; *Hsd17b11*, *P* = 0.028; *Hsp90b*, *P* = 0.01). **D** Western blotting showing up-regulation of BIP and CHOP in *Nfic*[-/-] pancreata (*n* = 7/group) (two-tailed Mann–Whitney *U*-test; *BiP*, *P* = 0.0175; *CHOP*, *P* = 0.047). Bar graph with densitometric quantification of data. **E** IF analysis of BIP in wild type and *Nfic*[-/-] pancreata (*n* ≥ 7/group). Boxplot shows quantification of the BIP expression intensity in WT and *Nfic*[-/-] pancreata. Individual dots correspond to the average of at least 15 images for each pancreas, scale bar = 10 μm. **F** ChIP-qPCR showing binding of NFIC, but not NR5A2, to the promoters of *Hspa5/Bip*-1, *Ddit3* and *Hsp90aa1* (*n* = 4) (two-tailed Mann–Whitney *U*-test; *BiP*, *Chop* and *Hsp90aa1*,

*P* = 0.0286 in the NFIC ChIP). **G** 266-6 cells incubated for 24 h or 36 h with vehicle or increasing TM concentrations (10 nM, 1 nM, 0.1 nM). Data shows the up-regulation of BIP and the down-regulation of NFIC by 24/36 h and the down-regulation of NR5A2 by 24 h in TM-treated cells (one of three independent experiments is shown). **H** Up-regulation of BIP and CHOP in 266-6 cells treated with TM upon *Nfic* knockdown. Boxplot shows quantification of data (*n* = 3 replicates/group) [two-tailed Student *T*-test; *BiP* in Nfic sh#1, *P* = 0.002; in Nfic sh #2, *P* = 0.004; in Nfic sh #3, *P* = 0.005. *Chop* in Nfic sh #1, *P* = 0.0038; in Nfic sh #2, *P* = 0.002; in Nfic sh #3, *P* = 0.001]. **I** Reduced BIP and CHOP expression in control and NFIC-overexpressing 266-6 cells treated with TM [two-tailed Student *T*-test; 24 h: *BiP*, *P* = 0.01; *CHOP*, *P* = 0.02; 36 h: *BiP*, *P* = 0.04; *CHOP*, *P* = 0.03]. Boxplots show quantification of data (*n* = 3 replicates/group). *P* < 0.1 (#), *P* < 0.05 (*), *P* < 0.01 (**); two-tailed Mann–Whitney *U*-test to calculate the significance in all panels. Boxplots show the minimum, the maximum, the sample median, and the first and third quartiles. Source data are provided as a Source Data file.

the selective role of NFIC (Fig. 5H, I). Overall, these results indicate that NFIC regulates the ER stress response in pancreatic acinar cells.

### *Nfic* is required for recovery upon induction of pancreatic damage

Acute pancreatitis is associated with a down-regulation of TFs involved in acinar differentiation and the up-regulation of ER stress and the UPR[37,38]. Therefore, we posited that NFIC might be required for a homeostatic response. Upon induction of an acute caerulein pancreatitis (7 hourly doses) in wild type mice, *Nfic* mRNA levels decreased at early time points (8 h) and were gradually restored (Fig. 6A). A similar expression pattern was observed at the protein level (Fig. 6B, C). To investigate whether *Nfic* inactivation affects damage and/or regeneration, we induced an acute pancreatitis in control and knockout mice. Serum amylase levels were similar in both mouse strains in basal conditions (wild type: 3168 ± 431; KO: 2637 ± 645; *P* = 0.4) and 1 h (wild type: 26204 ± 5158; KO: 25770 ± 10274; *P* = 0.88) and 4 h (wild type: 36515 + 12637; KO: 47693 ± 9208; *P* = 0.34) after the last dose of caerulein. At the histological level, wild-type and *Nfic*[-/-] pancreata showed similar damage at early time points (1–24 h) but *Nfic*[-/-] pancreata showed more prominent damage (edema, leukocyte infiltration, multifocal ADM, and acinar vacuolization) at later time points, up to day 14 (Fig. 6D–H). IHC confirmed the presence, in *Nfic*[-/-] pancreata, of an increased number of cleaved caspase 3[+] cells at day 2, a higher number of CD45[+] and KRT19[+] acinar cells at day 5, and a higher number of Ki67[+] cells at days 5 and 14 (Fig. 6F, G and Supplementary Fig. 10). These changes were accompanied by up-regulation of *Ddit3/Chop* and *Hsp17b11* mRNA in *Nfic*[-/-] pancreata at 48 h and day 5 (Fig. 6H). These results indicate that NFIC is required for homeostatic recovery from pancreatic damage.

### NFIC restrains PDAC initiation

Pancreatitis sensitizes the pancreas to the oncogenic effects of mutant *Kras*. We first analyzed NFIC expression in murine PanINs and PDAC from *Ptf1a-Cre*[+/KI]*;KrasG12V*[+/KI] (KC) using IHC and found that it is down-regulated in both preneoplastic and tumor cells (Fig. 7A). Similar findings were made in samples from patients: we found significant down-regulation of *NFIC* mRNA in PDAC (*n* = 118) when compared to normal tissue (*n* = 13) (Fig. 7B)[39]. Using IHC, we found that NFIC is consistently down-regulated in PanINs of low (*n* = 56) and high (*n* = 34) grade and in a subset of PDAC samples (*n* = 43) (Fig. 7C right panel and D).

TFs involved in acinar differentiation have been shown to suppress tumor initiation in mice. Inactivation of *Nfic* in the context of the *Ptf1a-Cre*[+/KI]*;KrasG12V*[+/KI] alleles resulted in an increased number of PanINs (and of the relative area occupied by them) in 18–24-week-old mice (Fig. 7E, F). Altogether, these findings also support that NFIC reduces PDAC initiation.

## Discussion

### NFIC regulates the pancreatic acinar program

Acinar differentiation was long thought to be a "digital" process controlled by the master PTF1 complex. Increasing evidence supports an "analog" differentiation model whereby additional TFs are required for "completion" of this process. Among them are NR5A2[6,7], HNF1A[37], GATA6[4], MIST1[5,40], and XBP1[41]. Here, we show that NFIC, a ubiquitous TF, is an acinar regulator present in a complex with NR5A2. NFIC is not crucially required for pancreas organogenesis. However, its inactivation results in incomplete acinar maturation and, in adult mice, it plays an important role in ribosomal RNA maturation and the ER stress response. In addition, it is required for proper recovery from damage and suppression of PDAC initiation. We identified NFIC as an NR5A2 partner through co-binding in normal mouse pancreas but it remains to be determined whether both proteins interact directly. The conservation of spacing between NR5A2 and NFIC motifs among NR5A2 target genes supports transcriptional cooperation. NR5A2 ChIP-Seq data from embryonic and adult pancreas indicate that the role of NFIC is mainly in the latter, supporting its requirement for completion of acinar maturation and highlighting a functional role distinct from that of NR5A2. The transcriptional program driven by NFIC overlaps partially with that of the tissue-restricted PTF1A, NR5A2, and MIST1 factors, indicating that multiple TFs cooperate to activate acinar differentiation. However, inactivation of *Nfic* has milder effects than inactivation of *Ptf1a* or *Nr5a2*, likely because the latter act at earlier stages of pancreatic development. NFI proteins were first proposed to be involved in the regulation of ubiquitous genes but they can also regulate tissue-specific genes[15,20], including *CEL* in the mammary gland and *DSPP* in odontoblasts[20,42]. We show that NFIC also regulates the acinar program in the pancreas. The down-regulation of *CDH1* mRNA and protein observed in *Nfic*[-/-] pancreata extends previous reports on *CDH1* regulation by NFIC in epithelial tissues[20,42] and changes in multiple aspects of cell adhesion programs were revealed by GSEA in the RNA-Seq analysis. The pancreata of young *Nfic*[-/-] mice also showed increased acinar proliferation and infiltrating leukocytes, associated with an up-regulation of inflammatory transcripts. This phenotype is similar to that of *Nr5a2*[+/-26], *Hnf1a*[-/-37], *Gata6*[-/-4], and *Ptf1a*[-/-28] pancreata. Inflammatory pathways were up-regulated in *Nfic*[-/-] mice and a direct effect at the epithelial cell level is supported by an up-regulation of inflammatory genes upon *Nfic* knockdown in 266-6 cells. Motif analysis suggests the involvement of NF-kB, PPARγ:RXRA, and REL in these processes. These findings suggest that the activation of pro-inflammatory phenotypes in mice in which acinar cells fail to acquire normal maturation can result from both direct (NR5A2 and GATA6)[4,7,10,11] and indirect (PTF1A and NFIC) mechanisms, in agreement with the ChIP-Seq data available. The differences in Ki67 expression and CD45[+] cell infiltration were abolished in older mice, suggesting

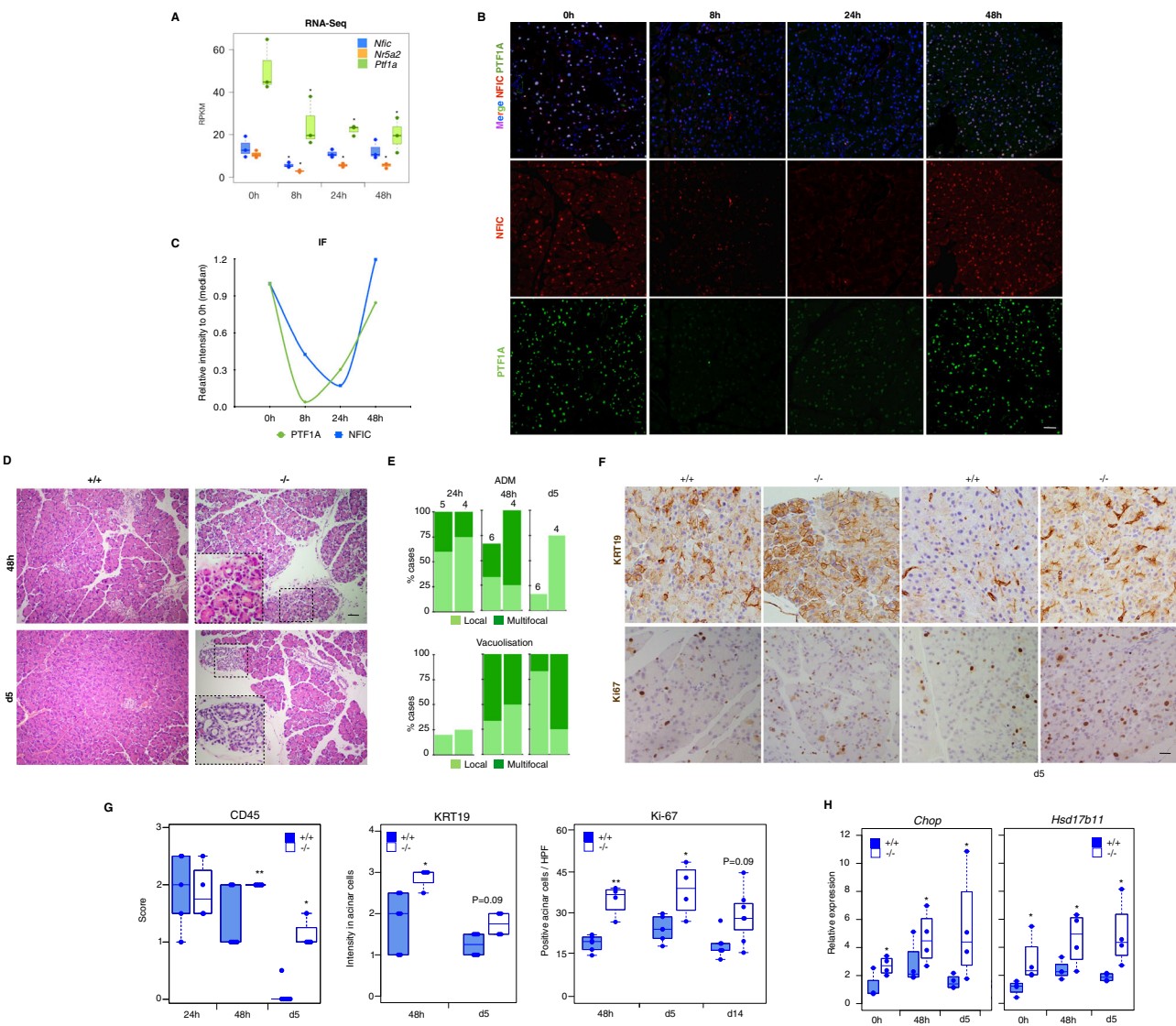

**Fig. 6 | NFIC is dynamically regulated during acute caerulein pancreatitis and is required for a homeostatic recovery. A** RNA-Seq analysis of *Nfic*, *Nr5a2*, and *Ptf1a* expression in wild-type mice upon induction of a mild acute pancreatitis (*n* = 3/ group). Significance was calculated compared to expression at 0 h (*P* < 0.05 (*) (*T*-test analogical method; *Nfic* at 8 h, *P* = 0.014; at 24 h, *P* = 0.45; at 48 h, *P* = 0.54; *Nr5a2* at 8 h, *P* = 0.023; at 24 h, *P* = 0.033; at 48 h, *P* = 0.036; *Ptf1a* at 8 h, *P* = 0.012; at 24 h, *P* = 0.027; at 48 h, *P* = 0.034). **B** IF analysis of NFIC and PTF1A upon pancreatitis induction showing NFIC down-regulation (*n* = 4/group, scale bar = 20 µm).
**C** Quantification of PTF1A⁺ and NFIC⁺ cells in wild-type mice during pancreatitis.
**D** Histological analysis of wild-type and *Nfic⁻/⁻* pancreata 48 h and 5 days after the induction of pancreatitis showing increased damage in mutant mice (*n* ≥ 4/condition, scale bar = 20 µm). **E** Pancreatitis scoring shows impaired recovery of *Nfic⁻/⁻* mice at 48 h and day 5 (*n* ≥ 4/condition). Damage was scored as (0–3) for each

parameter analyzed (*n* = 4/group). **F**, **G** IHC reveals increased expression of KRT19, a higher number of KI67⁺ acinar cells, and increased infiltration by CD45⁺ cells in *Nfic⁻/⁻* pancreata. Representative images are shown in (**F**), (scale bar = 10 µm); quantification of CD45, KRT19 and Ki67 expression (**G**) as described in Methods [*n* = 5/ group; *P* < 0.1 (#), *P* < 0.05 (*), *P* < 0.01 (**); two-tailed Mann–Whitney U-test. CD45: at 24 h, *P* = 1; at 48 h, *P* = 0.061; at day 5, *P* = 0.008). KRT19: at 48 h, *P* = 0.021; at day 5, *P* = 0.049. KI-67: at 24 h, *P* = 0.009; at 48 h, *P* = 0.038; at day 14, *P* = 0.059].
**H** RT-qPCR expression analysis showing up-regulation of *Ddit3* and *Hsd17b11* in *Nfic⁻/⁻* pancreata in basal conditions and after induction of an acute pancreatitis (*n* = 4 mice/group). *Chop:* at 0 h, *P* = 0.034; at 48 h, *P* = 0.046; at day 5, *P* = 0.057; *Hsd17b11:* at 0 h, *P* = 0.028; at 48 h, *P* = 0.057; at day 5, *P* = 0.028. Data are presented as mean values +/− SD. *P* < 0.05 (*). Source data are provided as a Source Data file.

organ adaptation, a phenotype that has not been described in other models to our knowledge.

## NFIC regulates expression of ribosomal genes and mitigates ER stress in the pancreas
A hallmark of acinar cells is their prominent capacity for protein synthesis, processing, and secretion[43]. This is achieved through acinar-specific transcriptional programs such as those driven by PTF1A[28], MIST1[40], and XBP1[41] and—as shown here—NFIC. Accordingly, a coordinated down-regulation of gene sets related to the digestive process and to protein synthesis/metabolism and oxidative phosphorylation

occurs in adult *Nfic⁻/⁻* pancreata. Interestingly, a highly significant defect in ribosomal RNA synthesis/transcription occurs in mutant mice. As previously described, the *Nfic* constitutive knockout mice used here display additional defects, including altered dental development and a shorter lifespan. However, several evidences support that our findings represent primary pancreatic epithelial defects rather than secondary effects involving other cell types: (i) reduced expression of digestive enzyme transcripts already occurs at late embryo stages, (ii) similar effects were observed in pancreatic tissue and in freshly isolated acini regarding digestive enzyme and ER stress phenotypes, (iii) these effects were reproduced upon *Nfic* knockdown in

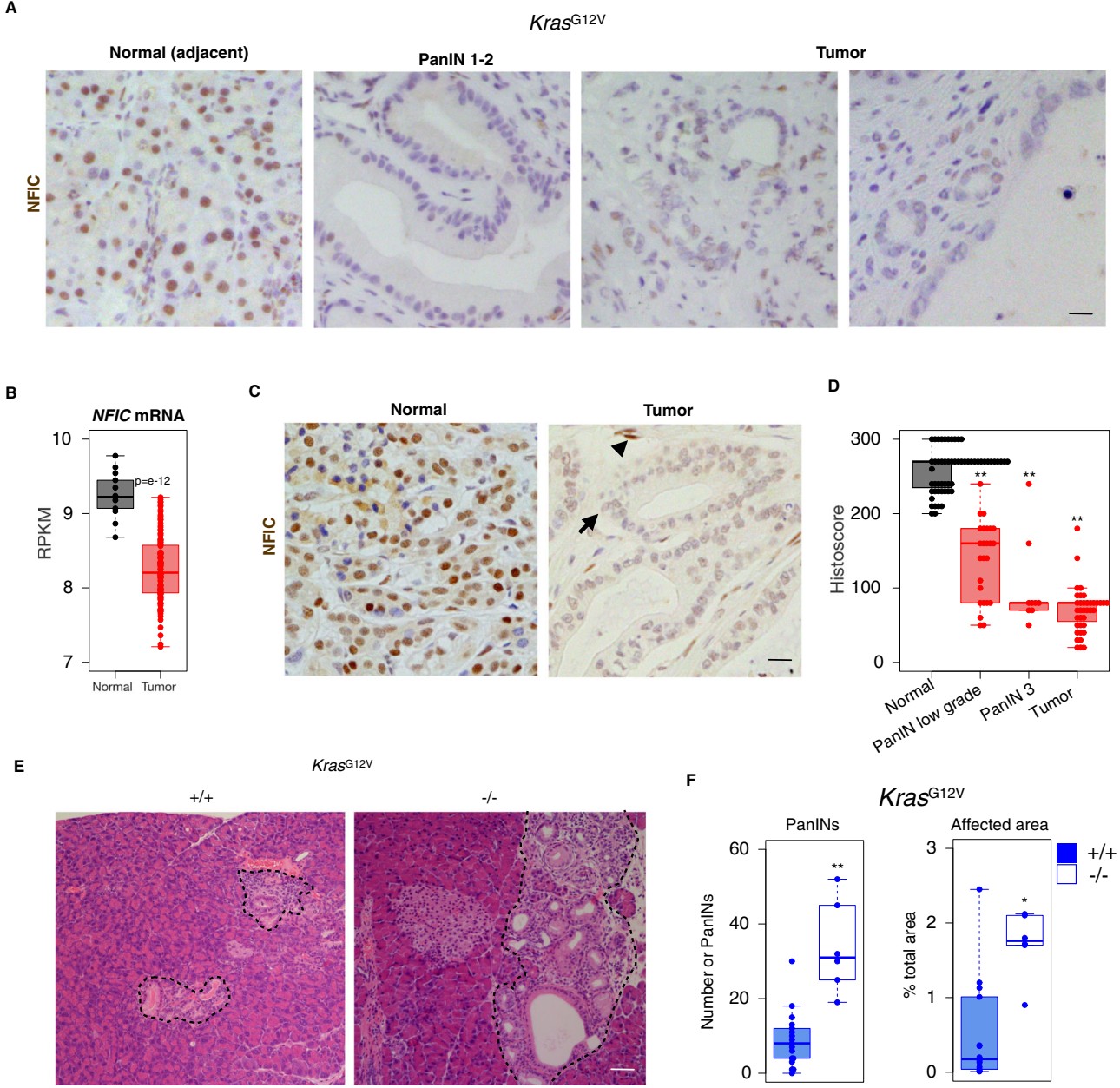

**Fig. 7 | NFIC restrains the formation of preneoplastic lesions in the pancreas.**
**A** IHC analysis of NFIC expression shows down-regulation in PanINs (right) and tumor cells compared to adjacent normal acinar cells (left) (one representative image of 5 replicates is shown, scale bar = 5 μm). **B** NFIC mRNA analysis of tumor samples and normal adjacent tissue assessed by microarrays[39] showing reduced expression in tumor samples (n = 14 for normal tissue and n = 118 for tumor, P = e-12, two-tailed Mann–Whitney U-test). **C**, **D** IHC analysis of NFIC expression in human PDAC specimens showing reduced expression in tumoral cells (arrow) compared to normal adjacent tissue or stromal cells (arrowheads) (n = 56 for normal; n = 25 for

PanIN1-2; n = 9 for PanIN-3; n = 43 for tumor; scale bar = 5 μm). Two-tailed Mann–Whitney U-test; PanIN1-2 vs. Normal, P = 1.666e-12; PanIN3 vs. Normal, P = 0.000004; Tumor vs. Normal, P = 2.2e-16. **E**, **F** Histological analysis of the pancreas of 14–20 week-old Kras^{G12V} or Kras; ^{G12V}Nfic^{-/-} mice showing increased number of PanINs and of the relative area occupied by pre-neoplastic lesions (n ≥ 6/genotype, scale bar = 20 μm). Two-tailed Mann–Whitney U-test; in **F** (# PanIN P = 0.0007; affected area P = 0.007). **D**, **F** Boxplots show the minimum, the maximum, the sample median, and the first and third quartiles. P < 0.05 (*), P < 0.01 (**). Source data are provided as a Source Data file.

266-cells. The generation of a Nfic conditional knockout mouse will contribute to further dissect these alterations and to assess the contribution of relevant signaling pathways (PI3K and mTOR) to these phenotypes. The importance of our findings in human disease is underlined by the fact that the transcriptomic differences between WT and Nfic^{-/-} were paralleled in normal human pancreata displaying low NFIC expression.

The high level of basal protein synthesis in acinar cells underlies constitutive activation of the UPR to reduce ER stress[38,44]. We observed an up-regulation of classical ER stress regulators, in the absence of an

increase in the ER compartment, and a tendency towards a reduced protein synthesis—likely contributing to the downregulation of UPR gene sets in Nfic^{-/-} pancreata. The finding that NFIC—but not NR5A2— binds the promoter of ER stress genes indicates a distinct role of the former in this process. This is supported by the changes in TM-mediated ER stress in 266-6 cells upon manipulation for NFIC gain-of-function and loss-of-function. We thus conclude that NFIC mitigates ER stress in acinar cells. Previous work has shown that NR5A2 is required for proper ER stress response in hepatocytes[36]. A re-analysis of published data shows that Nfic is down-regulated in Nr5a2^{-/-} hepatocytes in

basal conditions (fold change of −0.54)[36], suggesting that the deficient response to TM in *Nr5a2*[-/-] hepatocytes might partially occur through *Nfic* down-regulation. Indeed, NFIC is down-regulated upon TM-mediated ER stress in immortalized B cells[45].

## NFIC is dynamically regulated during pancreatitis and cancer

A failure to achieve complete acinar maturation is associated with more severe damage and delayed recovery during caerulein-mediated pancreatitis in several genetic mouse models including those where *Gata6*, *Mist1*, and *Ptf1a* are inactivated in the pancreas[4,46]. In addition, disruption of the UPR and ER stress responses induces acinar damage and can lead to acute or chronic pancreatitis[47]. We found that *Nfic* inactivation does not result in increased damage upon induction of a mild pancreatitis, but it impairs homeostatic recovery, with sustained up-regulation of *Ddit3/Chop* and *Hsp17b1* in *Nfic*[-/-] pancreata, indicating enhanced ER stress.

Several groups have shown that mild defects in the regulation of pancreatic transcriptional programs can sensitize to pancreatitis and that TFs act as tumor suppressors[10–12]. The role of NFIC in acinar cell differentiation and mitigation of ER stress suggested a contribution during tumorigenesis. NFIC has been proposed to act as a tumor suppressor in breast cancer, as it activates *TP53*, represses *CCND1* and *FOXF1*, and is down-regulated by *c-MYC* and *Ha-RAS* oncogenes[16–18,48,49]. In breast cancer, NFIC is down-regulated and high expression is associated with better prognosis[17]. Deregulation of other NFI family members has been reported in several tumor types: NFIB is overexpressed in metastatic neuroendocrine lung tumors and it drives metastatic progression of small cell lung cancers by increasing chromatin accessibility[50,51]. The lack of *NFIC* mutations and/ or genomic alterations in human PDAC (https://cancergenome.nih.gov/newsevents/newsannouncements/pancreatic_2017) suggests that other mechanisms may contribute to tumor development/progression.

Our work has some limitations. The use of a constitutive knockout mouse does not allow to disentangle the contribution of different cell types to the pancreatic phenotype, though we provide extensive evidence for an organ-autonomous effect. In addition, we cannot completely rule out a contribution of the dental and growth retardation phenotypes, which we have attempted to minimize by optimizing dietary care.

In summary, we show that NFIC belongs to a family of previously undescribed, ubiquitous, TF with tissue-specific functions in the pancreas that cooperates with NR5A2 to bind target genes and control their expression in vitro and in vivo. Unlike other pancreatic TF, the role of NFIC is restricted to the adult pancreas and distinctly affects RNA and protein metabolism and the UPR-mediated ER stress. Mutations leading to protein misfolding, activation of the UPR, and ER stress cause chronic pancreatitis and can contribute to the risk of PDAC[52], further supporting the role of NFIC in pancreatic homeostasis and disease.

## Methods

### Ethics statement

The experimental work reported in this manuscript complies with all relevant ethics regulations. The study protocol was approved by the Comité de Etica y Bienestar Animal, Instituto de Salud Carlos III (CBA 09_2015_v2) and the Comunidad Autónoma de Madrid (ES280790000186).

### Cells

HEK293 cells were obtained from the ATCC (CRL-1573.3); 266-6 cells were obtained from Dr. I. Rooman (VUB, Brussels, Belgium) who obtained them from ATCC (CRL-2151).

### Mice and experimental manipulations

The following mouse strains were used: *Nfic*[-/-][19], *Ptf1a*[+/Cre] knock-in[53], and *Kras*[G12V] conditional knock-in[8]. All crosses were maintained in a predominant C57BL/6 background. Experiments were performed using 8–25-week-old mice of both sexes, as indicated in the text, except for glucose tolerance tests where only males were used. Littermate mice were used as controls. Mice of both genotypes were co-housed. After weaning, mice were fed both with solid pellets and with pellets moistened with water or with a nutritionally fortified gel. The teeth of *Nfic*[-/-] mice were clipped using surgical scissors as deemed necessary by the animal room personnel, without intervention of the researchers, to avoid malocclusion. All animal procedures were approved by local and regional ethics committees (Institutional Animal Care and Use Committee and Ethics Committee for Research and Animal Welfare, Instituto de Salud Carlos III) and performed according to the European Union guidelines.

A mild acute pancreatitis was induced by 7 hourly injections of the cholecystokinin analog caerulein (Bachem) at 50 µg/kg. In brief, animals were weighed before the procedure and caerulein was administered intraperitoneally. Mice were killed by cervical dislocation after 1 and 4 h after the last caerulein injection or 24, 48 h, 5 and 14 days after the first caerulein injection. For the glucose tolerance test, male mice were fasted for 16 h and basal glycaemia was measured in tail blood. Mice received a glucose solution (2 g/kg) administered intraperitoneally and glycaemia was measured 15, 30, 60, and 120 min later using an automated glucose monitor (Accu-Chek® Aviva). Fasting glucose was considered as baseline (0 h). The number of mice used in each experiment is shown in the legend of each figure. For most experiments, ≥5 mice per group were used. No specific randomization method was used.

### Acinar cell isolation

Mice were sacrificed by cervical dislocation. The pancreas was injected with 2.5 mL of chilled collagenase P in HBSS (1.3 mg/mL), dissected, cut in small pieces, and incubated at 37 °C for 30 min (60U shaking speed). The reaction was stopped in a flow hood with 5% FBS/HBSS and the digest was centrifuged at 150 g for 2 min at 4 °C. After removing the supernatant, the pellet was washed twice with 5% FBS/HBSS, resuspended in 5% FBS/HBSS, filtered through a gauze, and rinsed with 5% FBS/HBSS. The resulting suspension was then pipetted through a 100 µm strainer, layered on a 30% FBS/HBSS solution, and centrifuged. The acinar cell fraction was suspended in RPMI supplemented with 10% FBS, Na pyruvate (1 mM) (Sigma-Aldrich), soybean trypsin inhibitor (STI) (Gibco) (0.1 mg/ml), and 1% Pen/Strep, and maintained at 37 °C for 24 h[26].

### Protein synthesis analysis by flow cytometry

Acinar cells were isolated as described above with minor modifications and maintained at 37 °C for 24 h in RPMI supplemented with 10% FBS, L-glutamine, Na pyruvate (1 mM), STI (0.1 mg/ml), and geneticin (25 mg/mL). Acinar cells were incubated with OP-P (20 mM) for 30 min. Cells were fixed in 4% PFA for 15 min at RT and labeled following a standard Click-It reaction protocol (Invitrogen, C10456). A Gallios Flow Cytometer (Beckman Coulter) was used to measure fluorescence; data were analyzed using FlowJo software. Samples without OP-P were used to determine background signal (control).

### Histology, immunofluorescence (IF) and immunohistochemical (IHC) analyses

Pancreata were immediately placed in buffered formalin or 4% paraformaldehyde. Histological processing was performed using standard procedures. To score damage in acute pancreatitis experiments, inflammation-related histological parameters [oedema, inflammatory cell infiltration, vacuolization, and acino-ductal metaplasia (ADM)] were scored blindly by IC and FXR according to the grade of severity (0–3).

IF and IHC analyses were performed using 3 µm sections of formalin-fixed paraffin-embedded tissues, unless otherwise indicated.

After deparaffinization and rehydration, antigen retrieval was performed by boiling in citrate buffer pH 6 for 10 min. For IF, sections were incubated for 45 min at room temperature with 3% BSA, 0.1% Triton X-100-PBS and then with the primary antibody overnight at 4 °C. For double or triple IF, the corresponding antibodies were added simultaneously and incubated overnight at 4 °C. Sections were then washed with 0.1% Triton–PBS, incubated with the appropriate fluorochrome-conjugated secondary antibody, and nuclei were counter-stained with DAPI. After washing with PBS, sections were mounted with Prolong Gold Antifade Reagent (Life Technology).

For IHC analyses, after antigen retrieval, endogenous peroxidase was inactivated with 3% $H_2O_2$ in methanol for 30 min at room temperature. Sections were incubated with 2% BSA-PBS for 1 h at room temperature, and then with the primary antibody overnight at 4 °C. After washing, the Envision secondary reagent (DAKO) was added for 40 min at room temperature and sections were washed x3 with PBS. 3,30-Diaminobenzidine tetrahydrochloride (DAB) was used as a chromogen. Sections were lightly counterstained with haematoxylin, dehydrated, and mounted. For some antibodies, an automated immunostaining platform was used (Ventana Discovery XT, Roche). A non-related IgG was used as a negative control. To validate the specificity of anti-NFIC antibodies, $Nfic^{-/-}$ pancreata were used as controls (Supplementary Fig. 1).

For CD45 quantification, whole digital slide images were acquired with an Axio Scan Z1, Zeiss scanner and then captured with the Zen Software (Zeiss). Image analysis and quantification were performed with the AxioVision software package (Zeiss). Briefly, areas of interest (AOI) were selected for quantification and then exported as individual TIFF images. CD45 staining were quantified using AxioVision 4.6 (Zeiss). Data obtained were then compiled and appropriately assessed. Images containing lymph nodes, and with artifactual staining or suboptimal cutting were eliminated from the analysis.

For quantification of KI67$^+$ positive cells, at least 10 random images from each pancreas were selected and only positive acinar cells were quantified. For BIP quantification, at least 10 random images from each pancreas were taken and fluorescence intensity was calculated using FIJI software (https://fiji.sc/). For semi-quantitative analysis of KRT19 staining, intensity was scored from 0 to 3 by IC. A list of the antibodies used for IHC and IF is provided in Supplementary Table 1.

## Quantitative RT-PCR (RT-qPCR)

For RNA isolation, pancreata were homogenized in denaturing buffer (4 M guanidine thiocyanate, 0.1 M Trizma HCl pH 7.5, 1% 2-mercaptoethanol) and processed as described earlier[26]. Total RNA was treated with DNase I (Ambion) for 30 min at 37 °C and cDNAs were prepared according to the manufacturer's specifications, using the TaqMan reverse transcription reagents (Applied Biosystems, Roche). RT-qPCR analysis was performed using the SYBR Green PCR master mix and an ABIPRISM 7900HT instrument (Applied Biosystems). Expression levels were normalized to endogenous $Hprt$ mRNA levels using the $\Delta\Delta C_t$ method. Minor modifications of the above method were used to assess rRNA maturation, including the use of $Gapdh$ mRNA for normalization. The results shown are representative of at least three biological replicates. The sequence of the primers used is provided in Supplementary Table 2.

## Immunoprecipitation and western blotting

Pancreata were snap-frozen for protein isolation. For immunoprecipitation of proteins from fresh total pancreas lysates, a piece of mouse pancreas was isolated and minced in 50 mM Tris-HCl pH 8, 150 mM NaCl, 5 mM EDTA, 0.5% NP-40 containing 3× phosphatase inhibitor cocktail (Sigma-Aldrich) and 3× EDTA-free complete protease inhibitor cocktail (Roche). Lysates were briefly sonicated until the protein solution was clear, cleared for 10 min at 12850 g at 4 °C and the supernatant was recovered. Antibody-coated protein A or protein G dynabeads (Life Technology) were used for immunoprecipitation. In brief, beads were washed three times with PBS and incubated with anti-NR5A2 or normal goat IgG (Millipore) overnight at 4 °C. After washing three times with PBS and twice with coupling buffer (27.3 mM sodium tetraborate, 72.7 mM boric acid), the dry beads were incubated overnight at 4 °C in freshly prepared 38 mM dimethyl pimelimidate dihydrochloride in 0.1 M sodium tetraborate. Afterwards, beads were washed three times with coupling buffer and once with 1 M Tris pH 9. Then, 1 ml of the Tris solution was added to the beads and incubated for 10 min at room temperature with rotation to block amino groups and stop crosslinking. Finally, beads were washed three times with storage buffer (6.5 mM sodium tetraborate/boric acid) and stored at 4 °C until used. Protein lysates (10-15 mg, tissues) were then incubated overnight at 4 °C with antibody-coated dynabeads (Thermo Fisher Scientific). Bound immune complexes were washed twice with lysis buffer containing NP-40, and then eluted by boiling in 2× Laemmli buffer (10% glycerol, 2% sodium dodecyl sulfate and 0.125 M Tris-HCl pH 6.8) for 5 min.

For western blotting, proteins were extracted from pancreatic tissue, isolated acinar cells or cultured cells using either Laemmli buffer, lysis buffer (50 mM Tris-HCl pH 8, 150 mM NaCl, 5 mM EDTA and 0.5% NP-40) or 5 M urea, supplemented with protease inhibitor and phosphatase inhibitor cocktails. Protein concentration was measured using the BCA reagent (Biorad), Nanodrop or extrapolated when using Laemmli lysis buffer. Proteins were resolved either by standard SDS-PAGE or 4-20% TGX pre-cast gels (Biorad) and transferred onto nitrocellulose membranes. A list of antibodies used for WB, ChIP and IP is provided in Supplementary Table 1. Densitometry analysis of digitalized western blotting images was performed using Fiji software (https://fiji.sc/).

## Chromatin immunoprecipitation (ChIP)

Pancreas tissue was minced, washed with cold PBS supplemented with 3× protease and phosphatase cocktail inhibitors, and then fixed with 1% formaldehyde for 20 min at room temperature. Glycine was added to a final concentration of 0.125 M for 5 min at room temperature. The fixed tissue was soaked in SDS buffer (50 mM Tris pH 8.1, 100 mM NaCl, 5 mM EDTA and 0.5% SDS) and homogenized using a douncer. The supernatant was collected after centrifugation and chromatin was sonicated with a Covaris instrument for 40 min (20% duty cycle; 10% intensity; 200 cycle), yielding DNA fragments with a bulk size of 300–500 bp. Samples were centrifuged to pellet cell debris. The amount of chromatin isolated was quantified using Nanodrop; an aliquot of this material was used as input for final quantification. Samples (0.5-1 mg of chromatin) were diluted with Triton buffer (100 mM Tris pH 8.6, 0.3% SDS, 1.7% Triton X-100 and 5 mM EDTA) to 1 ml and pre-cleared for 2 h with a mix of protein A and G (previously blocked with 5% BSA) at 4 °C. Antibody-coated beads were added: anti-NR5A2 (2 µg), anti-NFIC (1 µg), and rabbit anti-PTF1A serum (1/500). Non-related IgG was used as a control. After incubating for 3 h at 4 °C in a rotating platform, beads were successively washed with 1 ml of mixed micelle buffer (20 mM Tris pH 8.1, 150 mM NaCl, 5 mM EDTA, 5% w/v sucrose, 1% Triton X-100 and 0.2% SDS), buffer 500 (50 mM HEPES at pH 7.5, 0.1% w/v deoxycholic acid, 1% Triton X-100, 500 mM NaCl and 1 mM EDTA), LiCl detergent wash buffer (10 mM Tris at pH 8.0, 0.5% deoxycholic acid, 0.5% NP-40, 250 mM LiCl and 1 mM EDTA) and TE (pH 7.5), and then bound molecules were eluted by incubating overnight in elution buffer (containing 1% SDS and 100 mM NaHCO$_3$) at 65 °C, and treated with proteinase K solution (10 M EDTA, 40 mM Tris-HCl pH 6.5, 40 µg/ml proteinase K). The eluted DNA was purified by phenol–chloroform extraction. After isolation, pelleted DNA was resuspended in nuclease-free water (150 µl). Gene occupancy was then analysed by real-time PCR using 1 µl of the eluted DNA diluted in a final volume of 10 µl. The sequence of the primers used for ChIP-qPCR is provided in Supplementary Table 3.

### Nfic knockdown

NFIC expression was interfered in 266-6 cells using Mission shRNA lentiviral constructs purchased from Sigma-Aldrich. *Nfic* sh1 (TRCN0000374154 targeting ACAGACAGCCTCCACCTACTT), *Nfic* sh2 (TRCN0000310992 targeting TGTGTGCAGCCGCACCATATT), and *Nfic sh3* (TRCN0000301779, targeting GATGGACAAATCTCCATT CAA). Control cells were transformed using lentiviral particles transducing the scrambled vector CCGGCAACAAGATGA AGAGCACCAAC TCGAGTTGGTGCTCTTCATCTTGTTGTTTTT (shNT).

To produce lentiviral particles, HEK293-FT cells were allowed to reach 50% of confluence and transfected with 15 μg of shNT, *Nfic* sh1, *Nfic* sh2 or *Nfic sh3* plasmids together with 8 μg of psPAX and 2 μg of pCMV-VSVG helper plasmids using CaCl$_2$ 2 M HBSS. After 12 h, the supernatant was collected and replaced with 5 ml of fresh medium. The supernatant was collected 24 h, 48 h and 72 h after transfection. The medium was filtered (0.45 μm pore) and added to 266-6 cells (at 50–60% of confluence); 1 μg/mL of Polybrene (hexadimethrine bromide, Sigma-Aldrich 107689) was added to increase infection efficiency. After 2–3 rounds of infection, the supernatant was removed and replaced with fresh medium. One day later, puromycin (1–2 μg/ml) (Sigma-Aldrich) was added and 2 days later, the medium was replaced.

### NFIC lentiviral overexpression

*Nfic*-HA tagged cDNA was purchased from Addgene (https://www.addgene.org/31403/) and subcloned into the lentiviral vector pLVX-puro using *XhoI* and *XbaI*. Insert sequence was checked using enzymatic digestion and Sanger sequencing. The production of lentiviral particles and cellular infection were performed as described earlier. The medium from the transfectants was collected 24 h, 48 h and 72 h after transfection. Subsequently, 266-6 cells were infected using Polybrene as described earlier. After selection with puromycin for 24–48 h, resistant 266-6 cells were collected for RNA and protein analysis.

### Tunicamycin (TM) treatment

266-6 cells were seeded until they reached 70% confluence. After pilot dose-response experiments, a concentration of 10 nM was chosen; cells treated with TM or vehicle were collected at various time-points for RNA and protein analysis.

### RNA-Seq library preparation and analysis

RNA from wild type and *Nfic*$^{-/-}$ pancreata was isolated as described above. Average sample RNA Integrity Number was 8.4 (range 7.8–9.2), assayed on an Agilent 2100 Bioanalyzer. PolyA+ fraction was purified and randomly fragmented, converted to double stranded cDNA, and processed through subsequent enzymatic treatments of end-repair, dA-tailing, and ligation to adapters as in Illumina's "TruSeq Stranded mRNA Sample Preparation Part # 15031047 Rev. D" kit. Adapter-ligated library was completed by PCR with Illumina PE primers. The resulting purified cDNA library was applied to an Illumina flow cell for cluster generation and sequenced on an Illumina HiSeq2500 instrument by following manufacturer's protocols. Image analysis, per-cycle base calling and quality score assignment was performed with Illumina Real Time Analysis software. Conversion of Illumina BCL files to bam format was performed with the Illumina2bam tool (Wellcome Trust Sanger Institute−NPG). FASTQ sequencing files were mapped to the mm10 using STAR with default parameters. Quantification of raw transcripts counts and Transcripts Per kilobase Million (TPM) was performed using the *analyzeRepeats.pl* of HOMER. Differential expression analysis was produced using *getDiffExpression.pl* tool of HOMER[54]. Pathway analyses of differentially expressed genes were performed using Metascape[55] or molecular signature dataset of GSEA. The data containing the raw counts and TPM values for the genes in the downstream analyses, as well as the differentially expressed genes can be found in GEO (GSE126907). The Pearson correlation among samples was calculated from the expression value (expressed as fragments per kilobase of transcript per million mapped reads) of each gene for each sample by using the 'cor' command in R (https://www.r-project.org/). Principal component analysis was performed using the 'prcomp' command in R, from the correlation value of each sample.

RNA-seq data for *Nr5a2* (GSE34030), *Mist1* (GSE86288) mutant pancreata were downloaded from SRA. RNA-Seq of pancreata from wild type mice during pancreatitis was analysed as previously described[26] and is available under GSE84659.

Comparison of gene expression in normal human pancreata according to *NFIC* transcript levels was performed using RNA-Seq data from GTEX website (https://gtexportal.org/home/datasets, version 6) ($n = 171$)[56]. The expression data matrix was sorted by NFIC expression levels taking the 10 individuals scoring highest and lowest NFIC expression levels (*NFIC*$^{high}$, *NFIC*$^{low}$). Differential expression analysis using the DEGseq package of R (https://bioconductor.org/packages/release/bioc/html/DEGseq.html). MA-plot-based method with Random Sampling model -MARS[56] was applied and only genes with significance $P < 0.001$ were used in the analysis.

### Gene Set Enrichment Analysis (GSEA)

A ranking metric [−log10(*p*-value)/sign(log2FoldChange)] was used to generate a ranked gene list from the DEseq output. The list of pre-ranked genes was then analysed with using the molecular signature dataset of GSEA for Gene Ontology (GO), KEGG, REACTOME, HALLMARKS or CANONICAL PATHWAYS databases as described in the Figure legends and the text. Significantly enriched terms were identified using a false discovery rate (FDR) *q*-value of <0.25.

### NFIC ChIP-Seq and data processing

Chromatin from mouse pancreas tissue was extracted and processed as described above. For ChIP sequencing, libraries were prepared from purified DNA using "NEBNext Ultra II DNA Library Prep Kit for Illumina" from New England BioLabs (NEB, #E7645), as per the manufacturers' instructions. The resulting libraries were sequenced on Illumina HiSeq 2500, v4 Chemistry.

Data from NR5A2 ChIP-Seq in adult pancreata (SRR389293, SRR389294), NR5A2 ChIP-Seq in ES cells (GSM470523, GSM470524), PTF1A ChIP-Seq in adult pancreata (GSM2051452, GSM2051453), and MIST1 ChIP-Seq in adult pancreata (GSM2299654,GSM2299654, GSM2299655) were downloaded from the Gene Expression Omnibus website (https://www.ncbi.nlm.nih.gov/geo/) and analysis was performed as described[26]. Briefly, after the quality check by fastqc (v.0.9.4, Babraham Bioinformatics), the alignment and peak calling for the ChIP-seq data was performed using RUbioSeq+ pipeline[57]. Merging of replicate peaks and peak annotation was done using HOMER. Peak calling, annotation and motif enrichment was identified using HOMER (http://homer.ucsd.edu/homer/)[54]. Reads were directionally extended to 300 bp and, for each base pair in the genome, the number of overlapping sequence reads was determined and averaged over a 10-bp window to create a wig file to visualize the data in the University of California Santa Cruz (UCSC) genome browser.

NFIC ChIP-Seq data using GM12878, ECC1, HepG2, SK-N-SH and K562 cells were downloaded from (https://www.encodeproject.org/targets/NFIC-human/). ChIP-Seq peaks were analysed using Peak Analyser_1.4, using and Nearest Transcription Start Site parameter was used to annotate the genomic location of peaks. More than 95% of the target genes identified in the replicate with lowest number of target genes were included in the replicate with highest number. Among the two replicates, the one with highest number of identified target genes was taken: replicate 1 of NFIC ChIP-Seq in GM12878, NFIC ChIP-Seq in HepG2 and NFIC ChIP-Seq in SK-N-SH and replicate 2 of NFIC ChIP-Seq in ECC1 cell line.

## Other statistical analyses

Comparisons of quantitative data between groups was performed using one-sided Mann–Whitney *U*-test in all cases for which there was a prior hypothesis. For comparisons showing a normal distribution of data, two-tailed Student's *t*-test was used to calculate statistical significance. For comparison not showing a normal distribution, two-tailed Mann–Whitney *U*-test was used to calculate statistical significance. All group data are represented by the mean and errors bars are the standard deviation (SD). When comparing different groups within different variables, multiple comparison two-way ANOVA were used. Box plots represent the median and second and third quartiles (interquartile range, IQR) of the data. Error bars are generated by R software and represent the highest and lowest data within 1.5× IQR range. All statistical analyses were performed with Excel, R software, https://ccb-compute2.cs.uni-saarland.de/wtest/ or https://www.medcalc.org/calc/comparison_of_proportions.php. The random list of genes were generated using https://www.dcode.fr/random-selection and http://www.molbiotools.com/randomgenesetgenerator.html websites. Duplicated transcripts in RNA-Seq data were deleted for analysis. Dotted line refer to threshold for statistical significance ($-\log10[0.25] = 0.60$) or ($-\log10[0.05] = 1.30$). The statistical test used in each experiment is provided in the Figure legends or in the Source Data file.

## Reporting summary

Further information on research design is available in the Nature Portfolio Reporting Summary linked to this article.

## Data availability

All data are available in the main article, supplementary information, and source data or upon reasonable request to the authors. RNA sequencing data have been deposited in GEO with accession number GSE126907 and NFIC ChIP sequencing data have been deposited in GEO with accession number GSE181098. Source data are provided with this paper.

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

## Acknowledgements

We thank A. Efeyan, N. Djouder, and P. Martinelli for valuable discus-sions; M. Soengas and B. Bréant for providing antibodies; J. Perales and F. Al-Shahrour for help handling the GTEX dataset; R.M. Gronostajski and M. Barbacid, and C.W. Wright for providing mice; M. Barba, C. Yolanda, S. Rueda, L. Rodríguez, and T. Lobato for excellent technical assistance and help with animal care; and the Core Facilities and Bioinformatics Unit of Spanish National Cancer Research Center (CNIO) for support. This work was supported, in part, by grants SAF2011-29530, SAF2015-70553-R, RTI2018-101071-B-I00 from Ministerio de Ciencia, Innovación y Uni-versidades (Madrid, Spain) (co-funded by the ERDF-EU) and RTICC from Instituto de Salud Carlos III (RD12/0036/0034) to FXR and grants PID2020-119533GB-I00, from Ministerio de Ciencia, Innovación y Uni-versidades (Madrid, Spain), and GCB15152955MÉND, from the Spanish Association Against Cancer (AECC), to RM. I.C. was recipient of a Beca de Formación del Personal Investigador from Ministerio de Economía y Competitividad (Madrid, Spain). The research leading to these results has received funding from People Programme (Marie Curie Actions) of the European Union's Seventh Framework Programme (FP7/2007-2013) (REA grant agreement n° 608765"). S.P. was supported by a Juan de la Cierva Programme fellowship from Ministerio de Ciencia, Innovación y Universidades. C.B. is supported by a FPI Fellowship from Ministerio de Ciencia, Innovación y Universidades. I.M. was supported by a Fellowship from Fundació Bancària La Caixa (ID 100010434) (grant number LCF/BQ/ES18/11670009). CNIO is supported by Ministerio de Ciencia, Inno-vación y Universidades as a Centro de Excelencia Severo Ochoa SEV-2015-0510.

## Author contributions

I.C.: study concept and design; acquisition of data; analysis and inter-pretation of data; statistical analysis; drafting of the manuscript; S.P.: acquisition of data; analysis and interpretation of data; drafting of the manuscript; C.B.: acquisition of data; analysis and interpretation of data; drafting of the manuscript; I.F.: acquisition of data; analysis and inter-pretation of data; drafting of the manuscript; J.M.-A.: acquisition of data; analysis and interpretation of data; drafting of the manuscript. A.T.: acquisition of data; analysis and interpretation of data; J.M.-V.: analysis and interpretation of data; M.M.: acquisition of data; analysis and inter-pretation of data; F.G.: acquisition of data; analysis and interpretation of data; I.M.: analysis and interpretation of data; N.d.P.: technical support and acquisition of data; J.-C.P.: material support; R.J.M.: critical revision of the data and important intellectual content; J.M.: acquisition of data; analysis and interpretation of data; R.M.: acquisition of data; analysis and interpretation of data; obtained funding; F.X.R.: study concept and design; drafting of the manuscript; overall study supervision; obtained funding. All authors provided input about manuscript content.

## Competing interests

The authors declare no competing interests.
