## [Peer Review File · Nature Communications]

NFIC regulates ribosomal biology and ER stress in pancreatic acinar cells and restrains PDAC initiationREVIEWER COMMENTS

Reviewer #1 (Remarks to the Author):

Cobo and colleagues evaluated the implication of NFIC in the differentiation of pancreatic acinar cells.

By performing bioinformatics analyses on Chip-seq data from 4 pancreas-specific transcription factors (TF), authors identified NFIC as an important co-factor. Authors showed that NFIC is highly expressed in adult pancreas, more specifically in acinar cells and to a lesser extent in islets cells. Reduction or suppression of NFIC expression, by shRNA in 266-6 cells, by KO in mouse pancreas, or selection human samples with low NFIC, is associated with reduction of acinar specific enzymes such as CEL, CPA, CELA1/2 as well as PTF1, a key TF for acinar cells homeostasis. NFIC appears to cooperate with PTF1, NR5A2 or MIST1 to maintain acinar cells specific transcription program. Susceptibility to pancreatic damage and cancer formation of NFIC null mouse model were also assessed through caerulein-mediated acute pancreatitis and by crossing with Kras mutant mice, respectively. Overall, phenotypic analyses identify lack of NFIC as an enhancer of pancreatic damage and inflammation as well as neoplastic lesions formation, without inducing pancreatic cancer per se. Authors have also analyzed ribosomal programs and some aspects of UPR pathway in NFIC null acinar cells, suggesting that protein synthesis is also impacted in NFIC KO pancreas. Overall, this manuscript identified a new player in the acinar cells maturation, which cooperates with well-described pancreas-specific transcription factors. Experiments are well-designed and data are, in general, clear and convincing. This is especially true for pancreas histology and functions as well as for transcription factor activity, as expected for work from Dr Real lab. Nonetheless, sections describing the impact of NFIC on ribosomal components (rRNA and small subunit) or ER stress regulation provide poor demonstrations to support manuscript title. Overall, article quality would be greatly improved by performing experiments on those two sections and by clarifying some specific points, including the potential bias induced by NFIC KO model, which harbors teeth formation defects and growth delay.

Major points:

1. NFIC null mice suffer from teeth abnormalities and growth retardation. Without appropriate foods, NFIC mice will die prematurely due to feeding difficulties (ref 18). Defaults in nutrients availability (especially amino acids) have been shown to reduce both protein synthesis capacity, UPR/ISR and mTOR pathway in mouse pancreas (work from JA Williams and CD Logsdon). The present work is actually describing similar findings in NFIC KO pancreas. Although authors state that NFIC^{-/-} have a normal weight at 8 weeks, they did not provide details about housing, food type, etc. Thus, in order to support conclusions raised by authors, I would strongly recommend to perform experiments on freshly isolated acinar cells (as in Sup Fig 5 B) for which media composition is not subjected to changes.

2. Ribosomes assembly and functions and UPR have to be explore more in depth to support conclusions raised from Fig 4 and 5. Specifically, decrease of 5.8S rRNA (Fig 4A) or reduction ribosomal proteins from 40S subunit (RPS- Sup Fig 6B) is not sufficient to infer changes in ribosomes assembly or protein synthesis. In order to conclude about modification of ribosomes formation and function, authors should address maturation of rRNA (not using an antibody) and protein synthesis capacity in the absence of NFIC. Protein synthesis could be measure in vivo (Goodman FASEB 2011), but again, could be sensitive to fluctuation in food intake and amino acids availability (1rst comment). Thus, analysis of protein synthesis by 35S-Methionine and puromycin would have to be performed on NFIC KO acinar cells. In order, to address defects in ribosomes assembly (suggested by reduced RPS abundance), it seems appropriate to study polysomes formation in cells lacking NFIC. Signaling pathways regulating mRNA translation should be more carefully analyzed (see details below).

3. Unfolded protein response is a critical pathway for exocrine pancreas homeostasis. Deletion of PERK, ATF4 or eIF2 α phosphorylation site lead to defect in acinar cells formation or maintenance. In the pancreas, UPR is permanently activated to support to the large secretory function of this organ. Seminal work from JA Williams has also shown food intake or amino acids availability impacts on UPR signaling. Consequently, BiP and CHOP expression have to be monitored upon fasting (over 8hrs) and after re-feeding. Considering NFIC KO mouse phenotype, I would suggest to perform a detailed time course experiment, on freshly isolated acinar cells, with tunicamycin

and Caerulein to analyze the UPR activation (not limited to BiP and CHOP) including PERK, IRE1 and eIF2 α phosphorylation, ATF4, XBP1s.

4. Analysis of acute pancreatitis could be improved. Analysis of serum amylase activity along the time course of pancreatitis will provide additional evidence of pancreatic damage in NFIC null mice, especially in early phase 1 and 4h (described in material & methods). In the same line, H&E analysis should be performed at early time point. In fact, expression of pro-inflammatory cytokines and presence of CD45+ cells suggest an inflammatory status of NFIC KO pancreas, further supporter by the presence of local oedema (visible in Sup Fig 5 D). Thus, it appears critical to evaluate the presence of important oedema or focal ADM in older NFIC KO animals (4 to 6 months). In the same line, what is the impact of NFIC KO on survival of KRas PDAC mouse model?

Minor points:

Line 79: S6K1 and eIF4E are miswritten, please use official gene symbols (the same applies to RPS6, BiP, etc)

Line 114: remove "recent" work, data are over 9 years old

Page 6: description and figure numbers do not match (line 163 Fig 1D, line 165 Fig 1E, line 172 Fig 1F, etc)

Line 171: A reference is needed to state "well-validated antibody"

Line 199: MNK1 expression is not displayed in figure 2A.

Line 214-215: Measurement of proliferation of acinar cells by Ki67 is often biased in the presence CD45+ cells, which are also Ki67+ (Supp Fig 5 D-F). Authors should provide evidence that CD45+ were exclude from Ki67+ cells. The same applies to Fig 5F-G.

Line 217 to 224: Please provide statistics for this paragraph

Line 230: Enrichment of NFIC peaks is 20.66% on Fig 3B.

Line 260: Measuring 5.8S rRNA abundance with an antibody is quite unusual, especially considering the important processing of pre-ribosomal 47S rRNA. Molecular biology technics would be more appropriate and would draw robust conclusions about 5.8S, 18S and 28S rRNA abundance.

Line 264 to 273: Although mTOR pathway controls some key points of translation initiation and elongation, the inverse correlation is not necessarily true. This statement should be removed. In addition, S6K1 and RPS6 phosphorylation is regulated by mTOR activity, but eIF4E phosphorylation is rather connected to ERK (downstream of Ras) or P38 MAPK, independently of mTOR activity. The corresponding sentence should be edited accordingly. mTOR being a central sensor of nutrient availability, NFIC KO pancreatic extract could be problematic for signaling analysis (1st major point). As described above, fasting mice or using isolated acinar cells will be more accurate and complementary. For Fig 4C, expression of total RPS6, S6K1, EIF4E and ERK is required to raise proper conclusions.

Line 308: Phospho-S6 is not displayed on Fig 5F, but likely CHOP.

Line 341 to 343: PanCuRx dataset is not described and data are not shown. Please edit or remove this sentence.

Figure 5D: Relative intensity of WT are not normalized to 1. How are signals normalized?

Figure 7C: Display is incorrect.

Reviewer #2 (Remarks to the Author):

The studies in this manuscript center on the role of the transcription factor NFIC in acinar differentiation and pancreatic ductal adenocarcinoma. Cobo et al. showed that Nfic deletion in mice increases Kras-driven pre-neoplastic lesions. Moreover, NIFC is required proper recovery after pancreatitis. However, the authors just observed the alternations in the expression of genes relevant to acinar differentiation, inflammation, the mTORC1 pathway, ribosome, ER stress, and UPR. Most experiments described in this manuscript took an indirect approach to investigate how NFIC exerts its functions in acinar differentiation. The authors showed that NIFC directly interacts with NR5A2, but CHIP-seq data demonstrated that peaks bound by NFIC were identified with enrichment of motifs corresponding to transcription factors, which do not include NR5A2, indicating that NFIC does not work with NR5A2 for acinar cell differentiation. Further, these studies fall short of determining the mechanisms underlying how NFIC stimulates acinar cell differentiation, recovers pancreatic damages, and suppresses PDAC. I have several concerns below.

Major concerns:

1. Although NR5A2 was shown to associate with NFIC, the CHIP-seq of NFIC did not identify the motifs corresponding to NR5A2. Furthermore, to support the data of interaction between NFIC and NR5A2 (Fig. 1C), NFIC knockout pancreas should be used for the IP experiment.
2. The author described that "Nfic^{-/-} mice are viable and have a normal weight"; however, Nfic knockout mice show growth retardation and increased mortality under a standard diet condition (PMID: 12529411, PMID: 20729551, PMID: 32759468). Furthermore, NFIC mRNA is highly expressed in the ovary, prostate, fat, adrenal, skin, lung, kidney, liver, stomach, spleen compared to the pancreas (PMID 24309898). Would you employ the more direct approach to investigate the role of NFIC in acinar cell differentiation, such as tissue-specific deletion with Ptf1a-cre?
3. The IP experiments showed that NR5a2 was identified as an NFIC binding partner. However, several data from omic analyses, such as CHIP-seq (Fig. 3) and promoter scanning analysis (Supplementary Fig. 5), demonstrated no link of NFIC to NR5a2 in the pancreas. Would PPAR and NFkB be the binding partner to modulate inflammation and differentiation in acinar cells?
4. The author focused on the mTORC1 pathway and ribosomal proteins, but GSEA did not identify these pathways. Expression of only a few ribosomal genes is listed in Fig. 4 and Supplementary Fig. 6. Could you show the expression of all ribosomal genes and quantify the global translational efficiency in the pancreas using polysome profiling or another in vivo labeling method? Furthermore, although IF analysis showed 5.8S rRNA down-regulation in acinar cells, but to determine relative amounts of rRNA, could you show the image containing acinar, endocrine, and ductal cells like Supplementary Fig 1 and 2. Is the rRNA expression affected only in acinar cells but not other cells?
5. Ser at 209 of eIF4E is phosphorylated by MNK1/2 kinase that is downstream of the MAPK pathway. However, p-ERK levels are inversely correlated with p-eIF4E levels, demonstrating the inconsistency of the study.
6. Gene set enrichment analysis (GSEA) identified oxidative phosphorylation, focal adhesion, actin cytoskeleton, and the chemokine signaling pathway. However, pathways relevant to acinar differentiation, the mTORC1 pathway, and ER stress are not identified at the top of the list. These data demonstrated that NFIC is involved in mitochondrial functions and the actin cytoskeleton.
7. Could you measure insulin levels during GTT?

Minor concerns:

1. "Expression of PTF1A, CTRB1, and CPA proteins was similarly reduced (Figure 1H), suggesting an important role for NFIC in the regulation of late stages of acinar differentiation." and "Nfic^{-/-} pancreata appeared histologically normal and we did not find major differences in the expression of INS1, KRT19, and SOX9 (Supplementary Figure 3A), indicating that NFIC is not crucially required for pancreas development or differentiation." are not consistent.
2. The numbers in Figure 1 are not correct in the text.
3. The promoter assay should be done to determine the effect of NFIC on the promoter activities of target genes.
4. Would you determine the protein levels of PTF1A in the pancreas of WT and KO in Fig 2B.

Reviewer #3 (Remarks to the Author):

The manuscript by Cobo et al. characterizes the role of the transcription factor NFIC in pancreatic homeostasis, pancreatitis and pancreatic carcinogenesis. NFIC is identified as a pivotal regulator of

acinar differentiation and maintenance. Combined RNA- and ChIP seq studies in conditional transgenic pancreas models provide valuable insights into NFIC-dependent gene regulation and link the transcription factor to induction of acinar gene programs and reduction of inflammation and ER-stress programs. Upon pancreatitis induction, *Nfic* loss hampers pancreatic recovery and the transcription factor is down-regulated in murine and human pancreatic cancer precursor lesions and invasive pancreatic cancer.

The role of NFIC in the pancreas has not been studied so far. Hence, the data provided here are of high novelty and importance for the pancreas community. The experiments are – until unless stated differently, well controlled and interpreted and sufficient credit is given to statistics. However, as described below, there are some major aspects of the manuscript where the data is described in a very superficial way (in particular with regard to the mouse model and the NGS data acquisition). These points should be addressed in full to match the journal's standards. Further, additional experiments (e.g. using acinar cell extracts and in vitro ADM assays) should be performed to support the herein outlined role of NFIC-dependent gene regulation in the pancreas.

Major points:

- Given the detected PTF1A- and NR5A2 occupancy on the NFIC promoter: do these TFs activate NFIC transcription?
 - The characterization of the *Nfic*^{-/-} mice feels incomplete. HE stainings from 8-10 weeks old *Nfic*^{-/-} mice and older time points should be included. Do the mice develop acinar to ductal metaplasia when they grow older and do they develop a more inflammatory phenotype? A longer follow-up at least of a subset of mice would be appreciated. Further, it remains unclear, what *+/+* (Supl. 3) stands for? Are these wt mice or p48Cre mice (with the latter being the right control)? Can the authors please confirm the absence of NFIC expression in the *Nfcl*^{-/-} mice, e.g. via NFIC IF stainings which seem to work very convincing?
 - Is amylase expression in vivo altered (as suggested by the RNA-seq data)? Does *Nfic* depletion have an impact on the size of the overall acinar cell compartment? Differences in the composition of the acinar/ductal/endocrine compartment would probably be also reflected in the bulk RNA-seq. Hence, it would be important to exclude, whether the reduced expression of acinar genes upon *Nfic* loss represent a consequence of less acinar cells reflected in the RNAseq.
 - Substantial information and controls regarding the RNA-/ChIP-seq in *Nfic*^{-/-} mice are missing: how many mice/genotype have been utilized? Have RNA- and ChIP-seq been performed in the same mice? Please also display PCA plots or comparable controls representing similarities of replicates and differences between the strains. A heatmap showing the up-/down-regulation of genes (RNA-seq data) in the different mice/replicates would be highly appreciated.
 - Supl. 5B: can the authors please demonstrate the expression of some acinar markers in isolated acinar cells from *Nfic* wt and *-/-* mice? Considering the pancreatitis data of Fig. 6: Does the loss of *Nfic*^{-/-} foster a metaplastic phenotype when the acinar cells are cultured in collagen?
 - Please provide expression data +/- NFIC for those genes depicted in the ChIP-seq profiles of Fig. 3I.
 - Are ductal/inflammatory genes which have been identified to be up-regulated upon *Nfic*-deficiency also up-regulated upon shRNA-mediated TF depletion in 266.6 cells?
 - Pancreatitis induction (Fig. 6F). Could the authors please provide pictures on ADM containing regions?
 - Is the potential of *Nfic*^{-/-} mice to regenerate completely blocked or is the regeneration impaired and hence delayed? How does the *Nfic* pancreas look 7 or 10 days after pancreatitis induction? Further, the results part suggests that *Nfic*^{-/-} and control mice show comparable damage upon pancreatitis induction and refers to Fig. 6D. However, Fig. 6D does not show representative stainings of early time points (<48h). However, this would be important to assess, whether *Nfic*^{-/-} affects the damage/severeness of pancreatitis or the regeneration. Both conclusions would be interesting, but point towards different involvement of NFIC.
- Minor:
- Please revise the labeling in Figure 1: the NR5A2 ChIP seq data in E17.5 pancreas and mouse ES cells (should be Fig. 1B) is missing (please add). Subsequent sub figures in Fig.1 are not labeled correctly.
 - For sake of completeness, densitometry depicted in Fig. 2C should include NR5A2
 - Fig. 7D: x axis labelling is missing

Point by point responses to Reviewers' comments

We appreciate the thorough review of the paper and the suggestions, most of which have been addressed in our response. We believe that the revised version that is now being submitted is much improved.

Reviewer #1:

Major points

Cobo and colleagues evaluated the implication of NFIC in the differentiation of pancreatic acinar cells. By performing bioinformatics analyses on Chip-seq data from 4 pancreas-specific transcription factors (TF), authors identified NFIC as an important co-factor. Authors showed that NFIC is highly expressed in adult pancreas, more specifically in acinar cells and to a lesser extent in islets cells. Reduction or suppression of NFIC expression, by shRNA in 266-6 cells, by KO in mouse pancreas, or selection human samples with low NFIC, is associated with reduction of acinar specific enzymes such as CEL, CPA, CELA1/2 as well as PTF1, a key TF for acinar cells homeostasis. NFIC appears to cooperate with PTF1, NR5A2 or MIST1 to maintain acinar cells specific transcription program. Susceptibility to pancreatic damage and cancer formation of NFIC null mouse model were also assessed through caerulein-mediated acute pancreatitis and by crossing with Kras mutant mice, respectively. Overall, phenotypic analyses identify lack of NFIC as an enhancer of pancreatic damage and inflammation as well as neoplastic lesions formation, without inducing pancreatic cancer per se. Authors have also analyzed ribosomal programs and some aspects of UPR pathway in NFIC null acinar cells, suggesting that protein synthesis is also impacted in NFIC KO pancreas. Overall, this manuscript identified a new player in the acinar cells maturation, which cooperates with well-described pancreas-specific transcription factors. Experiments are well-designed and data are, in general, clear and convincing. This is especially true for pancreas histology and functions as well as for transcription factor activity, as expected for work from Dr Real lab. Nonetheless, sections describing the impact of NFIC on ribosomal components (rRNA and small subunit) or ER stress regulation provide poor demonstrations to support manuscript title. Overall, article quality would be greatly improved by performing experiments on those two sections and by clarifying some specific points, including the potential bias induced by NFIC KO model, which harbors teeth formation defects and growth delay.

We thank Reviewer #1 for the thoughtful review of our manuscript, the positive comments regarding its potential significance, and the constructive suggestions to address the impact of the dental phenotype of Nfic KO mice on the observations we report in the exocrine pancreas. Our conclusion on the role of NFIC in pancreatic acinar cell biology is based not only on the data provided in the original manuscript but also on several other lines of evidence that are now included in this revised version.

1. NFIC null mice suffer from teeth abnormalities and growth retardation. Without appropriate foods, NFIC mice will die prematurely due to feeding difficulties (ref 18). Defaults in nutrients availability (especially amino acids) have been shown to reduce both protein synthesis capacity, UPR/ISR and mTOR pathway in mouse pancreas (work from JAWilliams and CD Logsdon). The present work is describing similar findings in NFIC KO pancreas. Although authors state that NFIC^{-/-} have a normal weight at 8 weeks, they did not provide details about housing, food type, etc. Thus, in order to support conclusions raised by authors, I would strongly recommend performing experiments on freshly isolated acinar cells (as in Sup Fig 5 B) for which media composition is not subjected to changes.

We thank the referee for these important considerations. Indeed, NFIC is a crucial factor for tooth formation and we now describe in greater detail in the Methods section how wild type and Nfic KO mice were handled in our animal facility. All the experiments were designed to mitigate the impact of the dental phenotype on our analyses and dietary manipulations were similarly applied to wild type and Nfic KO mice, as they were co-housed. The text included in the Methods section is cut-pasted below:

"Mice of both genotypes were co-housed. After weaning, mice were fed both with solid pellets and with pellets moistened with water or with a nutritionally fortified gel. The teeth of *Nfic*^{-/-} mice were clipped using surgical scissors as deemed necessary by the animal room personnel, without intervention of the researchers, to avoid malocclusion."

Our conclusion that these findings reported result from a role of NFIC in pancreatic acinar cells is based not only on the data provided in the original manuscript but also on several other lines of evidence that were not included in the initial version of the manuscript because we felt they were redundant. However, given the concerns raised by the referee, and to provide more extensive information to the readers, we have revised the manuscript to include these data as well as the results of other experiments, as follows, and have commented in the Discussion:

- The pancreas of E17.5 *Nfic* KO embryos shows downregulation of *Cela2a*, *Pnlip*, and *Ctrb1*. These results are shown below and in Supplementary Figure 6D of the revised manuscript.

- Acinar cells isolated from 8-10 week-old *Nfic* KO mice show significantly reduced expression of acinar markers *Amy2b*, *Ctrb1*, *Cpa* and a significant upregulation of multiple ER stress/UPR markers, including spliced *Xbp1*, *Hspa5*, *Chop*, *Hsd17b11*, and *Hsp90b1*. These results are now included in Figure 2F and Figure 5C of the revised manuscript.

Ribosomes assembly and functions and UPR have to be explore more in depth to support conclusions raised from Fig 4 and 5. Specifically, decrease of 5.8S rRNA (Fig 4A) or reduction ribosomal proteins from 40S subunit (RPS- Sup Fig 6B) is not sufficient to infer changes in ribosomes assembly or protein synthesis. In order to conclude about modification of ribosomes formation and function, authors should address maturation of rRNA (not using an antibody) and protein synthesis capacity in the absence of NFIC. Protein synthesis could be measure in vivo (Goodman FASEB 2011), but again, could be sensitive to fluctuation in food intake and amino acids availability (1st comment). Thus, analysis of protein synthesis by 35S-Methionine and puromycin would have to be performed on NFIC KO acinar cells. In order, to address defects in ribosomes assembly (suggested by reduced RPS abundance), it seems appropriate to study polysomes formation in cells lacking NFIC. Signaling pathways regulating mRNA translation should be more carefully analyzed (see details below).

We acknowledge that an in-depth experimental characterization of the protein synthesis phenotype is required to reach robust conclusions. To this end, we have performed important additional experiments as summarized below:

- We have performed *Nfic* knockdown, using siRNA, in 266-6 cells, leading to reduced abundance of 5.8S rRNA, as shown below. The results of this experiment are mentioned in page 10 but we consider that they do not add much to the manuscript and suggest to leave them for reviewers' consideration only.

- The analysis of rRNA synthesis/maturation shows a dramatically reduced expression of 45S rRNA, mature 18S rRNA, 5.8S rRNA, and 28S rRNA in the pancreas of *Nfic* KO mice. These results are now shown below and in Figure 4B of the revised manuscript.

- We have specifically analyzed protein synthesis in acinar cells isolated from wild type and *Nfic* KO mice using flow cytometry. This well-established assay is often applied using fluorescence microscopy. However, this is not a good option for primary acinar cells given that they adhere poorly to the substrate and tend to produce aggregates (thus hampering adequate quantification at the individual cell level). Consequently, we have adapted this method for flow cytometry, making this non-radioactive assay a sensitive approach to assess protein synthesis with individual cell resolution. The results of these analyses, showing a non-significant trend towards reduced protein synthesis in KO cells ($p=0.13$), are shown below and included in Supplementary Figure 8D of the revised manuscript.

2. Unfolded protein response is a critical pathway for exocrine pancreas homeostasis. Deletion of PERK, ATF4 or eIF2 α phosphorylation site lead to defect in acinar cells formation or maintenance. In the pancreas, UPR is permanently activated to support to the large secretory function of this organ. Seminal work from JA Williams has also shown food intake or amino acids availability impacts on UPR signaling. Consequently, BiP and CHOP expression have to be monitored upon fasting (over 8hrs) and after re-feeding. Considering NFIC KO mouse phenotype, I would suggest to perform a detailed time course experiment, on freshly isolated acinar cells, with tunicamycin and Caerulein to analyze the UPR activation (not limited to BiP and CHOP) including PERK, IRE1 and eIF2 α phosphorylation, ATF4, XBP1s.

We appreciate the relevance of the experiments proposed by the referee. We have made extensive attempts to provide a rigorous answer to this point. We have performed 4-time course experiments for each treatment (4 μ g/ml tunicamycin or 10nM caerulein; 0, 1, 4 and 24h) and have used RT-qPCR and/or western blotting to assess expression of the genes/proteins suggested. Acinar cells isolated from WT pancreas displayed a normal appearance whereas there was clear evidence of stress in acini from Nfic KO pancreas. We have used 4 different methods (Trizol, Phenol-chloroform, Qiagen and Promega kits) to extract RNA from fresh acini but the quality of RNA was insufficient in some of the cases, particularly with the Nfic KO acini. Despite these technical problems, we can draw some interesting conclusions. Briefly, we find higher levels of BiP, Chop, and Ire1 α transcripts in basal conditions in Nfic KO acini; we also find higher levels of these transcripts in Nfic KO acini upon Tunicamycin treatment. In addition, we observe an increase in expression of BiP and Chop transcripts in Nfic KO acini upon caerulein treatment. The results are shown below (we do not include a statistical analysis because the data are based on selected experiments where the quality of the RNA was adequate and we did not always have 3 independent biological replicates).

Overall, the results are consistent with our conclusions on the role of NFIC in acinar cells. However, we do not feel confident enough to include them in the manuscript. We hope that the referee acknowledges the effort placed, the difficulties encountered, and our view that the paper should only contain those results that are sufficiently robust to merit going to print.

3. Analysis of acute pancreatitis could be improved. Analysis of serum amylase activity along the time course of pancreatitis will provide additional evidence of pancreatic damage in NFIC null mice, especially in early phase 1 and 4h (described in material & methods). In the same line, H&E analysis should be performed at early time point. In fact, expression of pro-inflammatory cytokines and presence of CD45+ cells suggest an inflammatory status of

NFIC KO pancreas, further supporter by the presence of local oedema (visible in Sup Fig 5 D). Thus, it appears critical to evaluate the presence of important oedema or focal ADM in older NFIC KO animals (4 to 6 months).

Again, the experiments suggested by the referee would substantially contribute to strengthen our conclusions. Moreover, we agree that a more detailed description of the histological phenotype upon induction of pancreatitis would be a valuable addition, including analyses in older mice. To address these issues, we have performed the following experiments:

- Analysis of serum amylase levels in wt and Nfic KO mice in basal conditions and 1h and 4h after the induction of pancreatitis. These experiments show no significant differences between both mouse strains neither in basal conditions nor upon caerulein-induced pancreatitis. The results are shown below and in page 12 of the revised manuscript.

- Histopathological evaluation of the pancreas of 20-25 week-old Nfic KO mice reveals no significant differences in edema, vacuolization, ADM, and lipomatosis. CDH1 expression, to reveal epithelial cells, was reduced in KO mice. We did not find differences in cell proliferation (measured by Ki67 immunostaining and quantification) or leukocyte infiltration (measured by CD45 immunostaining and quantification). These findings confirm the modest phenotype observed in basal conditions and suggest the participation of adaptive responses; the most relevant results are shown in Supplementary Figure 5B of the revised manuscript.

- To acquire a more detailed profile of the recovery from pancreatitis induction, we have extended the analysis of mice in which an acute 7-hour pancreatitis was induced up to day 14. The results indicate a persistently altered phenotype in KO mice at late time points, evidenced by slightly higher oedema and ADM scores and a significantly higher number of Ki67+ acinar cells. The results are shown in Figure 6G of the revised manuscript.

In the same line, what is the impact of NFIC KO on survival of KRas PDAC mouse model?

We completely agree with the referee that this an interesting and important question. However, it cannot be addressed with the mouse model used here because Nfic KO mice don't do well after 30 weeks age and - unless additional mutations are introduced to mutant Kras - it is not possible to properly assess differences in survival. Therefore, addressing this question will require the development of a conditional tissue-specific Nfic KO mouse, something which is beyond the scope of the present work.

Minor points

We thank the reviewer for a very thoughtful evaluation of the work. We have attempted to respond to the minor issues raised, as indicated below.

Line 79: S6K1 and eIF4E are miswritten, please use official gene symbols (the same applies to RPS6, BiP, etc)

Thanks for the thoughtful review of the nomenclature used in the paper.

Line 114: remove "recent" work, data are over 9 years old

The word "recent" has been removed from this sentence.

Page 6: description and figure numbers do not match (line 163 Fig 1D, line 165 Fig 1E, line 172 Fig 1F, etc)

We have carefully reviewed figure numbers and description to ensure the appropriate referencing of the text to the figures.

Line 171: A reference is needed to state “well-validated antibody”.

Thanks for raising this very important point, to which we have dedicated substantial effort as we are aware of the relevance of using antibodies of the appropriate specificity. In Supplementary Figure 1 of the revised manuscript, we provide an immunostaining of NFIC in wt and Nfic KO pancreata showing lack of staining in the latter. A sentence has now been added in page 7.

Line 199: MNK1 expression is not displayed in figure 2A.

Thanks for making us aware of this inconsistency. We have decided to remove this information from the manuscript as it does not add significantly to the findings reported.

Line 214-215: Measurement of proliferation of acinar cells by Ki67 is often biased in the presence CD45+ cells, which are also Ki67+ (Supp Fig 5 D-F). Authors should provide evidence that CD45+ were excluded from Ki67+ cells. The same applies to Fig 5F-G.

We appreciate the comment of the referee. We feel that we have a long track record of expertise in identifying acinar cells as per our prior publications (e.g., Molero et al. Gut, ref. 36) and we have carefully reviewed our digitalized images and confirmed the validity of the findings. We have now specified in the Methods section that the analysis of Ki67+ cells is restricted to acinar cells. We consider that it is necessary to provide images of our strategy but would be glad to do so if the referee considers that this is important.

Line 217 to 224: Please provide statistics for this paragraph

The statistical analysis of data related to experiments in Lines 217-224 of the old manuscript is now included.

Line 230: Enrichment of NFIC peaks is 20.66% on Fig 3B.

We performed these analyses using two different bioinformatics approaches which, as expected, yielded similar, but not identical, results. To avoid confusion, we are now reporting only the findings of one of them.

Line 260: Measuring 5.8S rRNA abundance with an antibody is quite unusual, especially considering the important processing of pre-ribosomal 47S rRNA. Molecular biology techniques would be more appropriate and would draw robust conclusions about 5.8S, 18S and 28S rRNA abundance.

This important point has already been discussed above and the new data are now shown in Figure 4B of the revised manuscript.

Line 264 to 273: Although mTOR pathway controls some key points of translation initiation and

elongation, the inverse correlation is not necessarily true. This statement should be removed.
Thanks for this remark; the statement has been removed.

In addition, S6K1 and RPS6 phosphorylation is regulated by mTOR activity, but eIF4E phosphorylation is rather connected to ERK (downstream of Ras) or P38 MAPK, independently of mTOR activity. The corresponding sentence should be edited accordingly.
Please, see response below.

mTOR being a central sensor of nutrient availability, NFIC KO pancreatic extract could be problematic for signaling analysis (1st major point). As described above, fasting mice or using isolated acinar cells will be more accurate and complementary. For Fig 4C, expression of total RPS6, S6K1, EIF4E and ERK is required to raise proper conclusions.
We agree with the referee that a more extensive analysis would be required to draw proper conclusions and have preferred to remove the signaling studies.

Line 308: Phospho-S6 is not displayed on Fig 5F, but likely CHOP.
Phospho-S6 has been removed from the text referencing Figure 5F of the revised manuscript. Thanks for picking up this error.

Line 341 to 343: PanCuRx dataset is not described, and data are not shown. Please edit or remove this sentence.
Since the data of PanCuRx dataset does not add much to the major conclusions of this work, we have deleted it from the text.

Figure 5D: Relative intensity of WT are not normalized to 1. How are signals normalized?
We now explain in the Methods section that normalization of western blotting signals is relative to loading control and to WT mice.

Figure 7C: Display is incorrect.
Thanks for the remark; we apologize for the error and have modified the text to indicate that the text refers to the right panel of Figure 7C.

Reviewer #2

We thank the referee for the helpful comments which we have responded to below.

Major points.

The studies in this manuscript center on the role of the transcription factor NFIC in acinar differentiation and pancreatic ductal adenocarcinoma. Cobo et al. showed that *Nfic* deletion in mice increases Kras-driven pre-neoplastic lesions. Moreover, NFIC is required for proper recovery after pancreatitis. However, the authors just observed the alternations in the expression of genes relevant to acinar differentiation, inflammation, the mTORC1 pathway, ribosome, ER stress, and UPR. Most experiments described in this manuscript took an indirect approach to investigate how NFIC exerts its functions in acinar differentiation. The authors showed that NFIC directly interacts with NR5A2, but CHIP-seq data demonstrated that peaks bound by NFIC were identified with enrichment of motifs corresponding to transcription factors, which do not include NR5A2, indicating that NFIC does not work with NR5A2 for acinar cell differentiation. Further, these studies fall short of determining the mechanisms underlying how NFIC stimulates acinar cell differentiation, recovers pancreatic damages, and suppresses PDAC. I have several concerns below.

Although NR5A2 was shown to associate with NFIC, the CHIP-seq of NFIC did not identify the motifs corresponding to NR5A2. Furthermore, to support the data of interaction between NFIC and NR5A2 (Fig. 1C), NFIC knockout pancreas should be used for the IP experiment.

We have performed an in-depth analysis of the interaction between NR5A2 and NFIC to provide further evidence that NR5A2 and NFIC are part of the same complex and bind

together to regulate the acinar transcriptional program. Specifically:

- The previous bioinformatics analysis was a *de novo* motif analysis. We now show HOMER motif analysis on the peak sequences from the NFIC CHIP-Seq data which is complementary to the *de novo* approach; we had not included this in the prior manuscript version for the sake of simplification but we agree that this should be included. The NR5A2 motif is found among of the top 10 most enriched motifs in the NFIC ChIP-Seq peaks. providing additional evidence on the functional cooperation between NFIC and NR5A2. The results are shown in Figure 3A of the revised manuscript.

de novo motifs	best match	% target	% backg.	$-\log_{10}$ (FDR)
	NFI	44.26	9.00	2741
	ATOH1	43.01	27.59	347
	NF1	18.99	8.96	307
	BORIS	6.28	1.50	277
	FOXM1	17.76	8.89	247
	NR5A2	14.04	7.02	190
	GATA	17.25	10.73	122
	HOXB13	4.05	1.41	107
	Sp2	17.32	11.24	104

- We have performed NR5A2 immunoprecipitation, followed by mass spectrometry of proteins in the immune complexes, and have found that peptides corresponding to the NFIC sequences are among the most significantly enriched. These findings are now shown in Figure 1C of the revised manuscript. For the sake of comprehensiveness, we have included all the information from the IP-MS experiment in Supplementary Table 1.

The author described that “Nfic^{-/-} mice are viable and have a normal weight”; however, Nfic knockout mice show growth retardation and increased mortality under a standard diet condition (PMID: 12529411, PMID: 20729551, PMID: 32759468). Furthermore, NFIC mRNA is highly expressed in the ovary, prostate, fat, adrenal, skin, lung, kidney, liver, stomach, spleen compared to the pancreas (PMID 24309898). Would you employ the more direct approach to investigate the role of NFIC in acinar cell differentiation, such as tissue-specific deletion with Ptf1a-cre?

We completely agree with the reviewer that using a constitutive Nfic KO mouse imposes limitations on the interpretation of the results, given the fact that NFIC is expressed in other tissues/cell types that could contribute to the pancreatic phenotype. To address the potential impact of the dental defect of Nfic KO mice on the observations made in the exocrine pancreas, we now describe in greater detail how wild type and Nfic KO mice were handled in our animal facility. All experiments were designed to mitigate the impact of the dental phenotype on our analyses. Dietary manipulations were similarly applied to wild type and

Nfic KO mice as they were co-housed. Our conclusion on role of NFIC in pancreatic acinar cell biology is based not only on the data provided in the original manuscript but also on several other lines of evidence that are now included in this revised version. The relevance of NFIC specifically in pancreatic cells is addressed by the experiments performed using Nfic KO acinar cells, as well as those using 266-6 cells upon Nfic knockdown. However, we cannot completely rule out the contribution of other cell types in vivo, as we indicate in the Discussion as one of the limitations of our work (page 16). To the best of our knowledge, a conditional Nfic KO mouse model is not available and such experiments - while important - are beyond the scope of this work.

The IP experiments showed that NR5a2 was identified as an NFIC binding partner. However, several data from omic analyses, such as CHIP-seq (Fig. 3) and promoter scanning analysis (Supplementary Fig. 5), demonstrated no link of NFIC to NR5a2 in the pancreas. Would PPAR and NFkB be the binding partner to modulate inflammation and differentiation in acinar cells?

We thank the reviewer for raising this interesting point. We have now performed additional motif analyses using HOMER; these analyses provide more robust evidence of the link between NR5A2 and NFIC genomic binding. Figure 3A of the revised manuscript demonstrates the enrichment of NR5A2 motifs in NFIC ChIP-Seq peaks. In addition, Supplementary Figure 8E of the revised manuscript shows the presence of NFIC motifs in the promoter of genes that are overexpressed in Nfic KO pancreata. We believe that this analysis, together with the NR5A2 IP-Mass Spec experiment, strongly supports the notion that NR5A2 and NFIC are part of the same transcriptional regulatory complex.

Following the reviewer's suggestion, we have also checked the expression of Pparg and Nfkb1 in the pancreas of Nfic KO mice and found that both are upregulated (log2 fold change 1.38, FDR<0.05 for Pparg; and log2 fold change 0.56, FDR <0.05 for Nfkb1). We concur with the referee that they are likely candidates to contribute but we have not dealt with this aspect in sufficient depth and, therefore, have preferred to down-play this point in the paper. We believe that this an important point to explore in the future.

The author focused on the mTORC1 pathway and ribosomal proteins, but GSEA did not identify these pathways. Expression of only a few ribosomal genes is listed in Fig. 4 and Supplementary Fig. 6. Could you show the expression of all ribosomal genes and quantify the global translational efficiency in the pancreas using polysome profiling or another in vivo labeling method? Furthermore, although IF analysis showed 5.8S rRNA down-regulation in acinar cells, but to determine relative amounts of rRNA, could you show the image containing acinar, endocrine, and ductal cells like Supplementary Fig 1 and 2.

We acknowledge that a link between the alteration of ribosomal/protein production program and loss of NFIC would strengthen our results. We have complemented our initial GSEA analysis using Metascape and the findings are now shown in Figure 5A.

We also agree that it is important to analyze the cell autonomous contribution of NFIC in

acinar cells to the phenotype observed in the KO pancreata but this will require generating a conditional KO mouse, which is beyond the scope of this paper. In addition, we agree that an *in vivo* labelling method to detect translation efficiency would be helpful to further support the notion that NFIC regulates ribosomal program of acinar cells, as also requested by reviewer 1. To achieve this, we have performed the following additional experiments and analyses:

- We have analyzed protein synthesis in acinar cells isolated from wild type and *Nfic* KO mice using flow cytometry. This well-established assay is often applied using fluorescence microscopy. However, this is not a good option for primary acinar cells given that they adhere poorly to the substrate and tend to produce aggregates (thus hampering adequate quantification at the individual cell level). Consequently, we have adapted this method for flow cytometry, making this non-radioactive assay a sensitive approach to assess protein synthesis with individual cell resolution. The results of these analyses, showing a non-significant trend towards reduced protein synthesis in KO cells ($p=0.13$), are shown below and included in Supplementary Figure 8D of the revised manuscript.

- We found that oxidative phosphorylation, macroautophagy, metabolism of amino acids, protein maturation and pancreatic secretion are among the most significantly downregulated gene sets in the *Nfic* KO pancreata (Figures 4D, 5A, and Supplementary Figure 8). *Nfic*-/- pancreata display higher levels of P62 expression as assessed by WB and IHC. These findings are also shown in Supplementary Figure 9D of the revised manuscript.

As explained in our response to comment 2 of Referee #1, experiments using *Nfic* sh RNA knock down in 266-6 cells show reduction of 5.8S rRNA staining, recapitulating the phenotype observed in acinar cells of the *Nfic* KO pancreata. This is now included in the text

in page 11 and shown below for reviewer's consideration. We feel that these results do not add sufficiently to the paper and would prefer to keep them for reviewer's consideration only.

Is the rRNA expression affected only in acinar cells but not other cells?

The intensity of immunostaining of 5.8S rRNA in islets is much lower than in acinar cells, as shown in the accompanying immunofluorescence staining. We feel that it is not necessary to show these data in the paper but the microphotograph is included for the reviewer's consideration and a comment to this issue has been added in page 10.

4. Ser at 209 of eIF4E is phosphorylated by MNK1/2 kinase that is downstream of the MAPK pathway. However, p-ERK levels are inversely correlated with p-eIF4E levels, demonstrating the inconsistency of the study.

We thank Referee #2 for raising this point. We feel that a more thorough analysis of signaling is required to fully understand how NFIC impacts on various pathways and have opted for deleting this part of the work.

Gene set enrichment analysis (GSEA) identified oxidative phosphorylation, focal adhesion, actin cytoskeleton, and the chemokine signaling pathway. However, pathways relevant to acinar differentiation, the mTORC1 pathway, and ER stress are not identified at the top of the list. These data demonstrated that NFIC is involved in mitochondrial functions and the actin cytoskeleton.

We thank the referee for raising this thoughtful point. As discussed above (comment 4). We have performed new gene set enrichment analysis using Metascape which shows that downregulated genes in Nfic KO pancreata are enriched in protein maturation and pancreas secretion gene sets. The acinar pathway is not well represented in standard gene sets: we have specifically used the acinar signature described by the group of R. MacDonald in Masui et al and show a highly significant down-regulation of acinar genes (Supplementary Figure 5A,B).

Could you measure insulin levels during GTT?

We agree with the referee that a potential endocrine defect in Nfic KO mice is of scientific interest. However, our work focuses on the role of NFIC in the exocrine pancreas. While measuring insulin during the GTT will certainly provide interesting information, assessing endocrine function in-depth would require more extensive studies that are not within the scope of this work.

Minor concerns:

1. “Expression of PTF1A, CTRB1, and CPA proteins was similarly reduced (Figure 1H), suggesting an important role for NFIC in the regulation of late stages of acinar differentiation.” and “Nfic^{-/-} pancreata appeared histologically normal and we did not find major differences in the expression of INS1, KRT19, and SOX9 (Supplementary Figure 3A), indicating that NFIC is not crucially required for pancreas development or differentiation.” are not consistent.

We respectfully disagree with the referee on this point. Expression of INS1, KRT19, and SOX9 is restricted to islet and/or ductal cells. Histology of all pancreatic compartments was normal in basal conditions. Our statement is that "NFIC is not crucially required for pancreas development or differentiation" and we think that the word "crucial" indicates that defects occur (in acinar cells, at a late stage because cell specification and the acinar program are largely preserved in KO mice) but they are subtle. Therefore, we would prefer to leave these sentences in the manuscript.

2. The numbers in Figure 1 are not correct in the text.

We thank the referee for the thoughtful review of the manuscript and have now corrected the text and the reference for Figure 1 accordingly.

3. The promoter assay should be done to determine the effect of NFIC on the promoter activities of target genes.

We thank the referee for bringing up this point. We have now performed luciferase reporter assays in HEK293T cells using a fragment of the Ela1b promoter - which contains putative binding sites for NFIC - and show increased reporter activity. These results are now shown as Figure 1J of the revised manuscript.

4. Would you determine the protein levels of PTF1A in the pancreas of WT and KO in Fig 2B.

We have analyzed PTF1A protein expression by IF (see Methods section) and found reduced expression in the Nfic KO pancreata. Considering the relatively modest differences in expression and the challenge of detecting PTF1A in total pancreatic lysates (the protein has a mobility that overlaps with that of digestive enzymes that are expressed at much higher levels), we felt that western blotting experiments would likely not add substantially novel information. These results are now shown as Figure 2D and 2E of the revised version of the manuscript.

Reviewer #3 (Remarks to the Author):

The manuscript by Cobo et al. characterizes the role of the transcription factor NFIC in pancreatic homeostasis, pancreatitis and pancreatic carcinogenesis. NFIC is identified as a pivotal regulator of acinar differentiation and maintenance. Combined RNA- and CHIP seq studies in conditional transgenic pancreas models provide valuable insights into NFIC-dependent gene regulation and link the transcription factor to induction of acinar gene programs and reduction of inflammation and ER-stress programs. Upon pancreatitis induction, Nfic loss hampers pancreatic recovery and the transcription factor is down-regulated in murine and human pancreatic cancer precursor lesions and invasive pancreatic cancer.

The role of NFIC in the pancreas has not been studied so far. Hence, the data provided here are of high novelty and importance for the pancreas community. The experiments are – until unless stated differently, well controlled and interpreted and sufficient credit is given to statistics. However, as described below, there are some major aspects of the manuscript where the data is described in a very superficial way (in particular with regard to the mouse model and the NGS data acquisition). These points should be addressed in full to match the journal's standards. Further, additional experiments (e.g. using acinar cell extracts and in vitro ADM assays) should be performed to support the herein outlined role of NFIC-dependent gene regulation in the pancreas.

Major points:

1. Given the detected PTF1A- and NR5A2 occupancy on the NFIC promoter: do these TFs activate NFIC transcription?

We appreciate that this is an interesting point. The analyses that we reported in the previous version of the paper strongly suggest that this is the case. This is substantiated by the new CHIP-Seq data included in this revised manuscript. The efficiency of co-transfection experiments in 266-6 cells is rather low and we have not been able to address this question properly. We hope that the referee shares with us that this experiment does not substantially change the conclusions of the paper.

2. The characterization of the Nfic^{-/-} mice feels incomplete. HE stainings from 8-10 weeks old Nfic^{-/-} mice and older time points should be included. Do the mice develop acinar to ductal metaplasia when they grow older and do they develop a more inflammatory phenotype? A longer follow-up at least of a subset of mice would be appreciated. Further, it remains unclear, what +/+ (Supl. 3) stands for? Are these wt mice or p48Cre mice (with the latter being the right control)?

We thank Reviewer #3 for raising these points. We have improved the analyses and have

characterized the pancreas of older (20-25 week-old) *Nfic* KO mice. Beyond this age, the mice do not do well and their phenotype may reflect broader systemic alterations not directly related to the role of NFIC in the pancreas. The new data are included as part of the response to Comment 4 of Reviewer #1:

- Analysis of serum amylase levels in wt and *Nfic* KO mice in basal conditions and 1h and 4h after the induction of pancreatitis. These experiments show no significant differences between both mouse strains in basal conditions and 1h and 4h after caerulein-induced pancreatitis. The results are shown below and in page 12 of the revised manuscript.

- Histopathological evaluation of the pancreas of 20-25 week-old *Nfic* KO mice reveals no significant differences in oedema, vacuolization, ADM, lipomatosis, cell proliferation (measured by Ki67 immunostaining and quantification), and leukocyte infiltration (measured by CD45 immunostaining and quantification). These findings confirm the modest phenotype observed in basal conditions in young mice; the results are shown and in Supplementary Figure 5 of the revised manuscript.

- In addition, we have made it clear that “+/+” means *Nfic* +/+ and that “-/-” means *Nfic* -/- and that we have used a constitutive KO mouse strain. Therefore, the proper controls are wild type mice, as used throughout our work. The use of p48-Cre mice would be inappropriate in this setting. We have added a sentence in the discussion about the interest of generating a conditional *Nfic* KO mouse.

3. Can the authors please confirm the absence of NFIC expression in the *Nfic*^{-/-} mice, e.g. via NFIC IF stainings which seem to work very convincing?

We appreciate the referee's request to include these data in the paper as they are highly important. We now provide an immunostaining of NFIC in wt and *Nfic* KO pancreata showing lack of staining in the latter (Supplementary Figure 1 of the revised manuscript). A sentence has now been added in page 7.

4. Is amylase expression in vivo altered (as suggested by the RNA-seq data)?

We showed in the prior version of the paper that the expression of several digestive enzymes in the pancreas of knockout mice is modestly reduced (old Figure 2). Following the referee's request, we have now analyzed amylase expression in tissue from wild type and *Nfic* KO mice and show that - as suggested by the RNA-seq data - amylase protein levels are lower. These findings are in agreement with the data shown in the western blot from Figure 2 and, therefore, we suggest to keep them only for referee's consideration.

5. Does Nfic depletion have an impact on the size of the overall acinar cell compartment? Differences in the composition of the acinar/ductal/endocrine compartment would probably be also reflected in the bulk RNA-seq. Hence, it would be important to exclude, whether the reduced expression of acinar genes upon Nfic loss represent a consequence of less acinar cells reflected in the RNAseq.

We thank the reviewer for raising this interesting point. First, we would like to point out that we now show new data indicating that the relative weight of the pancreas is significantly lower in Nfic KO mice ($P < 0.01$). To address the question related to the acinar cell compartment, we have digitalized H-E sections and quantified the exocrine components, finding no significant differences between mouse genotypes (WT = 96.98 ± 1.61 , KO = 94.46 ± 1.67 ; $P\text{-val} = 0.18$). This information has been added in page 7 of the manuscript. In addition, we have analyzed the expression of endocrine and ductal genes and found no differences in Nfic KO pancreata vs. WT. We consider that these data do not add significantly to the manuscript and, therefore, include them here only for reviewer evaluation.

	Log 2 fold change vs. WT	Adjusted P- value
Krt19	0.293	0.258
Sox9	0.215	0.604
Ins1	0.126	0.899
Gcg	1.032	0.265
Sst	0.386	0.708

6. Substantial information and controls regarding the RNA-/ChIP-seq in Nfic-/- mice are missing: how many mice/genotype have been utilized? Please also display PCA plots or comparable controls representing similarities of replicates and differences between the strains. A heatmap showing the up-/down-regulation of genes (RNA-seq data) in the different mice/replicates would be highly appreciated.

We thank the reviewer for raising these points. We have included the information on the number of mice/genotypes used and have performed the following additional analyses:

- Principal component analysis (left) and correlation analyses (right) of the samples used for the RNA-Seq. The results are shown in Figure 2B of the revised manuscript. In addition, we include below the correlation plot of the transcriptomes of the mice used for the RNA-Seq analyses, for referee's consideration only. If it is felt that the correlation plot should be included as Supplementary Material, we will be glad to do so.

Following the request of the referee, we have now included a heatmap for a subset of genes whose expression is downregulated (acinar genes; acinar transcription factors) or upregulated genes (right: inflammatory markers) in individual Nfic KO pancreata. These results are shown in Supplementary Figure 6A and 7C of the revised manuscript.

7. Have RNA- and ChIP-seq been performed in the same mice?

The ChIP-Seq and RNA-Seq experiments were performed in different sets of mice of similar age, housed following the same experimental conditions and handled similarly.

8. Supl. 5B: can the authors please demonstrate the expression of some acinar markers in isolated acinar cells from Nfic wt and -/- mice?

This issue was also raised by reviewer 1. We include the response offered above. Acinar cells isolated from 8-10 week-old Nfic KO mice show significantly reduced expression of acinar markers Amy2b, Ctrb1, Cpa and a significant upregulation of multiple ER stress/UPR markers, including spliced Xbp1, Hspa5, Chop, Hsd17b11, and Hsp90b1. The results are now included in Figure 2F and 5C of the revised manuscript.

9. Considering the pancreatitis data of Fig. 6: Does the loss of Nfic^{-/-} foster a metaplastic phenotype when the acinar cells are cultured in collagen?

We appreciate the referee's comment. However, we would like to say that we do not find these experiments very informative: acinar cells spontaneously undergo metaplastic changes in collagen (and in suspension or 2D cultures). In our modest opinion, this renders comparisons less valuable, despite that some authors have used them in the literature.

10. Please provide expression data +/- NFIC for those genes depicted in the ChIP-seq profiles of Fig. 3I.

Thanks for requesting this additional information. We have now clarified that the ChIP-Seq results shown correspond to genes that are downregulated (Figure 3I, top) or upregulated (Figure 3I, bottom) in the Nfic KO pancreata. We provide below a table with the changes in the expression of Ccla2a, Nr0b2, Bhlha15, Pparg, Cfi and Rara. The full dataset of differentially expressed genes is now included as Supplementary Table2 of the revised manuscript.

	Log2 fold change	Adjusted P-value
Ccla2a	-1.077	6.13E-05
Nr0b2	-0.868	0.0004
Bhlha15	-0.638	0.0001
Pparg	1.172	0.0007
Cfi	1.518	0.001
Rara	0.761	0.012

11. Are ductal/inflammatory genes which have been identified to be up-regulated upon Nfic-deficiency also up-regulated upon shRNA-mediated TF depletion in 266.6cells?

We have analyzed the expression of a subset of the genes up-regulated in the KO pancreas (Sox9, Krt19, Ccl5, Cxcl13) in 266-6 cells in which Nfic was knocked down with lenti-shRNA as compared to non-targeting (NT) control. The results show a similar up-regulation in cultured acinar cells, supporting the validity of the tissue analyses. These results are shown below and in Supplementary Figure 7D of the revised manuscript.

12. Pancreatitis induction (Fig. 6F). Could the authors please provide pictures on ADM-containing regions?

We have included microphotographs of H-E-stained sections from the pancreas of mice in which pancreatitis was induced, highlighting regions of ADM in Figure 6D of the revised manuscript.

13. is the potential of Nfic -/- mice to regenerate completely blocked or is the regeneration impaired and hence delayed? How does the Nfic pancreas look 7 or 10days after pancreatitis induction? Further, the results part suggests that Nfic-/- and control mice show comparable damage upon pancreatitis induction and refers to Fig.6D. However, Fig. 6D does not show representative stainings of early time points (<48h). However, this would be important to assess, whether Nfic-/- affects the damage/severeness of pancreatitis or the regeneration. Both conclusions would be interesting, but point towards different

involvement of NFIC.

Following the suggestion of this referee and of referee #1, we have now included a more detailed analysis of the pancreatitis phenotype at later timepoints. These data are now shown in Figure 6 of the revised manuscript.

Minor:

- Please revise the labeling in Figure 1: the NR5A2 ChIP seq data in E17.5 pancreas and mouse ES cells (should be Fig. 1B) is missing (please add). Subsequent sub figures in Fig.1 are not labeled correctly.

Thanks for the thorough review of the manuscript and apologies for the error. The labelling in Figure 1 has now been corrected.

- For sake of completeness, densitometry depicted in Fig. 2C should include NR5A2.

We have now added the quantification of the NR5A2 blot displayed in Figure 2C.

- Fig. 7D: x axis labelling is missing.

We thank the referee for picking up this omission which has now been fixed.

REVIEWER COMMENTS

Reviewer #1 (Remarks to the Author):

At first, I would like to thank the authors for carefully addressing each point raised in the initial review of this article. This has undoubtedly improved the overall quality and clarity of the manuscript.

My initial major concern about specific housing of NFIC KO animals facing the dental phenotype has been successfully addressed. Regarding the other questions, Cobo and colleagues have provided several experiments demonstrating the impact on NFIC KO on reducing expression of acinar markers at different stages of pancreatic development and on isolated acini (Fig 2). Similarly, authors confirmed the upregulation of UPR markers upon deletion of NFIC (Fig 5). In addition, authors gave strong evidences of downregulation of rRNA synthesis and maturation in NFIC null models (Fig 4).

I also appreciate the attempts to question the role NFIC on ER stress response. I trust that working with acinar cells is quite challenging, knowing the amount of RNase produced by these cells.

Despite these positive comments, several concerns remain to be solved:

1 - Deletion of NFIC is massively reducing ribosome biogenesis (Fig 4). Nevertheless, the functional consequences on protein synthesis capacity of NFIC KO acinar cells appear very mild (Sup Fig 8). Furthermore, during pancreatitis induction, pancreas has to produce large amount of enzymes including amylase, chymotrypsin,..., following each cerulein injection. Thus, it is overall surprising that the small pancreas of NFIC KO mice (Fig 2a) expressing fewer enzymes (Fig 2c- f) can induce a similar amount of circulating amylase (page 12) despite the reduced abundance of ribosomes. Altogether, ribosome abnormalities are quite descriptive with no (or reduced) correlation with other NFIC null pancreatic functions. Authors should provide an explanation for these observations. An attempt was made in the discussion "We observed an up-regulation of classical ER stress regulators and a tendency towards a reduced protein synthesis, likely contributing to the down-regulation of UPR gene sets in *Nfic*^{-/-} pancreata", but this doesn't really match with results as regulator (BIP) and effector (Chop) are both upregulated in NFIC KO pancreas. Thus, supporting this notion, and as suggested by Supp Fig9, it would be relevant to measure endoplasmic reticulum size in NFIC KO cells and pancreas through IF of ER-resident protein such as Calnexin or BIP.

2 - Experimental pancreatitis has a prolonged effect on NFIC KO pancreas both at histological and transcriptional levels. The sustained expression of CHOP in the absence of NFIC, under basal- (Fig 2B-D) or stressed-condition (Fig 6H), indicates a potential propensity of NFIC null cells to undergo apoptosis. This phenomena has been described in primary pulp cells of NFIC KO animals and participate to dental abnormality (PMID: 19386589). I would suggest to measure apoptosis in NFIC KO pancreas and acinar cells, under basal and/or pancreatitis settings (cleaved-PARP or caspases IHC, annexinV, etc). This quite simple experiment would provide supplemental evidences for NFIC role in the maintenance of pancreas homeostasis. In addition, it could explain the reduced size of the pancreas despite the elevated acinar cell proliferation (Ki67 staining).

3 - The title of the manuscript strongly states about the suppression of PDAC initiation by NFIC. Looking at figure 7, this is an overstatement. NFIC does not "suppress" the appearance of early lesion but likely delay, mitigate or restrain it as indicated in Figure 7 title. Thus, I would strongly suggest to change titles of the manuscript and corresponding paragraph together with the conclusion on PDAC initiation. I agree with authors that developing tissue-specific KO will be appropriate to study NFIC function in pancreatic cancer, but will be out of the scope of the current manuscript.

Minor points:

Protocol for RNAseq library generation for NFIC^{-/-} pancreas is missing, especially if ribo-depletion or polyA selection was performed. This is of particular importance considering the large variation

rRNA (which accounts for 80% of cellular RNA) in NFIC KO cells.
Page 12 line 398: a bracket is missing
Page 14 line 473: should be UPR activation.

Reviewer #2 (Remarks to the Author):

The concerns raised by the reviewers were partly addressed by the authors. However, major issues have not been solved yet.

1. The new data from HOMER motif analysis and IP-MS cannot directly support the conclusion. In the new analysis, NFIC does not still identify the motifs corresponding to NR5A2. To support the data of interaction between NFIC and NR5A2 (Fig. 1D), NFIC knockout pancreas should be used for the IP experiment as a negative control.
2. The authors cannot ignore the previous studies from other groups about Nifc knockout mice. Nifc knockout mice show growth retardation and increased mortality under a standard diet condition (PMID: 12529411, PMID: 20729551, PMID: 32759468).
3. The IF experiment is not suitable to quantify levels of proteins due to the lack of internal controls.
4. The author focused on the mTORC1 pathway, ribosomal proteins, and the ISR pathway, but the analysis added to the revised manuscript did not still identify these pathways.
5. The inconsistency between the levels of p-ERK and p-eIF4E is not addressed.

Reviewer #3 (Remarks to the Author):

Thank you for providing an extensive revision which has improved the quality of the manuscript. From my point of view, based on the revision work this manuscript is now suitable for publication.

Point-by-Point response to reviewers' comments

Reviewer #1 (Remarks to the Author):

At first, I would like to thank the authors for carefully addressing each points raised in the initial review of this article. This has undoubtedly improved the overall quality and clarity of the manuscript.

My initial major concern about specific housing of NFIC KO animals facing the dental phenotype has been successfully addressed. Regarding the other questions, Cobo and colleagues have provided several experiments demonstrating the impact on NFIC KO on reducing expression of acinar markers at different stages of pancreatic development and on isolated acini (Fig 2). Similarly, authors confirmed the upregulation of UPR markers upon deletion of NFIC (Fig 5). In addition, authors gave strong evidences of downregulation of rRNA synthesis and maturation in NFIC null models (Fig 4).

I also appreciate the attempts to question the role NFIC on ER stress response. I trust that working with acinar cells is quite challenging, knowing the amount of RNase produced by these cells.

We thank the reviewer for her/his appreciation of the effort placed in responding to the critiques.

Despite these positive comments, several concerns remain to be solved:

1 - Deletion of NFIC is massively reducing ribosome biogenesis (Fig 4). Nevertheless, the functional consequences on protein synthesis capacity of NFIC KO acinar cells appear very mild (Sup Fig 8). Furthermore, during pancreatitis induction, pancreas has to produce large amount of enzymes including amylase, chymotrypsin,..., following each cerulein injection. Thus, it is overall surprising that the small pancreas of NFIC KO mice (Fig 2a) expressing fewer enzymes (Fig 2c- f) can induce a similar amount of circulating amylase (page 12) despite the reduced abundance of ribosomes. Altogether, ribosome abnormalities are quite descriptive with no (or reduced) correlation with other NFIC null pancreatic functions. Authors should provide an explanation for these observations. An attempt was made in the discussion "We observed an up-regulation of classical ER stress regulators and a tendency towards a reduced protein synthesis, likely contributing to the down-regulation of UPR gene sets in Nfic-/- pancreata", but this doesn't really match with results as regulator (BIP) and effector (Chop) are both upregulated in NFIC KO pancreas. Thus, supporting this notion, and as suggested by Supp Fig9, it would relevant to measure endoplasmic reticulum size in NFIC KO cells and pancreas through IF of ER-resident protein such as Calnexin or BIP.

We concur with the reviewer that the effects at the level of ribosome biogenesis are dramatic while there is only a trend for reduced protein synthesis, as shown in Supplementary Figure 8D. We are making similar contrasting observations in other models our lab is working on which leads us to conclude that we still have a rather incomplete understanding of how ribosome biogenesis, protein synthesis, and ER biology work in acinar cells.

The possible paradox in NFIC KO mice regarding the amount of enzymes produced and the circulating amylase may not be so surprising considering that the serum amylase levels are influenced not only by the content in acinar cells but also by the rate of mis-secreted amylase (basolateral vs. apical/luminal).

Regarding the referee's suggestion, to assess ER size, we would like to point out that we already showed that the levels of BiP are higher in the pancreas of KO mice using western blotting (Fig. 5D) in the previous version of the ms. and we have validated this in an independent cohort of mice. Following her/his request, we also assessed expression of Calreticulin - an ER marker - and find that there are no differences between wild type and mutant mice (see below; results only for reviewer's perusal). These results suggest increased ER stress in the absence of expansion of the ER compartment. We have added a sentence in the discussion to highlight these points in page 14.

2 – Experimental pancreatitis has a prolonged effect on NFIC KO pancreas both at histological and transcriptional levels. The sustained expression of CHOP in the absence of NFIC, under basal- (Fig 2B-D) or stressed-condition (Fig 6H), indicates a potential propensity of NFIC null cells to undergo apoptosis. This phenomena has been described in primary pulp cells of NFIC KO animals and participate to dental abnormality (PMID: 19386589). I would suggest to measure apoptosis in NFIC KO pancreas and acinar cells, under basal and/or pancreatitis settings (cleaved-PARP or caspases IHC, annexinV, etc). This quite simple experiment would provide supplemental evidences for NFIC role in the maintenance of pancreas homeostasis. In addition, it could explain the reduced size of the pancreas despite the elevated acinar cell proliferation (Ki67 staining).

We appreciate this suggestion and have performed the requested experiments: we assessed apoptosis using cleaved caspase 3 staining in basal conditions and in pancreatitis samples: in basal conditions, there were no significant differences between wild type and knockout mice. Upon induction of pancreatitis, there was a significant increase in the proportion of apoptotic cells in knockout mice at day 2. These results are presented in pages 7 and 11 and shown in Supplementary Figures 5 and 10.

3 – The title of the manuscript strongly states about the suppression of PDAC initiation by NFIC. Looking at figure 7, this is an overstatement. NFIC does not “suppress” the appearance of early lesion but likely delay, mitigate or restrain it as indicated in Figure 7 title. Thus, I would strongly suggest to change titles of the manuscript and corresponding paragraph together with the conclusion on PDAC initiation. I agree with authors that developing tissue-specific KO will be appropriate to study NFIC function in pancreatic cancer, but will be out of the scope of the current manuscript.

Thanks for the suggestion which is, indeed, appropriate. We have changed the title to a more conservative statement. The new title is " NFIC regulates ribosomal biology and ER stress in pancreatic acinar cells and restrains PDAC initiation".

Minor points:

Protocol for RNAseq library generation for NFIC^{-/-} pancreas is missing, especially if ribo-depletion or polyA selection was performed. This is of particular importance considering the large variation rRNA (which accounts for 80% of cellular RNA) in NFIC KO cells.

We thank the reviewer for a very thoughtful evaluation of the work, including the methods. PolyA selection was performed to prepare the libraries. We have included the complete information in pages 20-21 of the revised manuscript.

Page 12 line 398: a bracket is missing

Page 14 line 473: should be UPR activation.

Thanks for pointing out these minor issues, which have been corrected in the new manuscript.

Reviewer #2 (Remarks to the Author):

The concerns raised by the reviewers were partly addressed by the authors. However, major issues have not been solved yet.

Thanks for appreciating our efforts to respond to the referee's suggestions.

1. The new data from HOMER motif analysis and IP-MS cannot directly support the conclusion. In the new analysis, NFIC does not still identify the motifs corresponding to NR5A2.

We modestly disagree with the referee. In the new analysis included in the prior version of the paper, using the random genomic sequences provided by HOMER as background, we can see the NR5A2 motif is one of the top six most significant motifs identified in the NFIC ChIP-Seq peaks (see below). We are attaching below panel A from Figure 3, which we refer to.

Regarding the IP-MS, if that is an additional issue, the NR5A2 IP-MS data clearly show a notable (2.7) fold-change enrichment of NFIC peptides with high statistical significance. In addition, the immunoprecipitation-western blotting experiments confirm the specificity of the interaction (see below).

To support the data of interaction between NFIC and NR5A2 (Fig. 1D), NFIC knockout pancreas should be used for the IP experiment as a negative control.

We appreciate that this experiment further complements the many evidences presented in our paper regarding the specificity of the NFIC antibody [i.e. immunohistochemistry (Suppl. Fig. 1) and western blotting using pancreas from *Nfic* knockout mice (Fig. 2D), knockdown in z66-6 cells (Fig. 5H)] and the specificity of the interaction (i.e. IP-western and IP-mass spectrometry). To provide further evidence, as requested by the referee, we have now performed the immunoprecipitation-western blotting experiment using pancreatic extracts from wild type and knockout mice. To do so, we have been forced to use a different anti-NFIC antibody since the one used most extensively in our prior work has been discontinued since our paper was submitted. As shown below, we provide evidence that NR5A2 immunoprecipitates of pancreatic lysates from wild type mice contain NFIC while this is not the case when using lysates from knockout pancreata. We have added a sentence indicating the results of this new experiment in page 6 and show the data in Supplementary Figure 1A of the new version of the manuscript.

2. The authors cannot ignore the previous studies from other groups about Nfic knockout mice. Nfic knockout mice show growth retardation and increased mortality under a standard diet condition (PMID: 12529411, PMID: 20729551, PMID: 32759468).

Thank you very much for these suggestions; we appreciate the concerns of this referee on this matter. We aim at being very careful and respectful of the published work: regarding the phenotype of Nfic knockout mice, we have provided extensive citations concerning the growth retardation phenotype and increased mortality of Nfic KO mice. In fact, one of the 3 references that this referee asked to include is already cited in the paper (PMID 12529411 is ref. 19, previously reference 18). Following the reviewer's suggestion, we have now added the other two references (PMID: 32759468, PMID: 20729551) as refs. 23 and 24 (cited in pages 5 and 15).

3. The IF experiment is not suitable to quantify levels of proteins due to the lack of internal controls.

We respectfully disagree, in part, with the reviewer. Proper quantification of protein expression probably requires mass spectrometry with internal standards. All other antibody-based methods have one or another bias, including western blotting. However, these methods - with their limitations - are almost universally used by the scientific community. Therefore, while acknowledging that they do not provide absolute quantification of protein expression, we think that they fall within the acceptable approaches used in cell biology research.

4. The author focused on the mTORC1 pathway, ribosomal proteins, and the ISR pathway, but the analysis added to the revised manuscript did not still identify these pathways.

We respectfully disagree with the referee. Figure 5A shows differential enrichment of several pathways related to ribosomal function and the integrated stress response: autophagy, amino acid metabolism, protein maturation, and response to starvation, among others. These pathways are related to mTOR function, that being the reason why we assessed this pathway in greater detail.

5. The inconsistency between the levels of p-ERK and p-eIF4E is not addressed.

As we indicated in the response to reviewers' comments to the second version, we agree with the referee that the signaling aspects of our work required more extensive analysis and we opted for excluding them from the paper until we have an improved understanding of the role of NFIC in acinar cells. Therefore, we feel that this comment is not pertinent to the work described in the most recent version submitted and in the current version of the paper.

Reviewer #3 (Remarks to the Author):

Thank you for providing an extensive revision which has improved the quality of the manuscript. From my point of view, based on the revision work this manuscript is now suitable for publication.

We thank the reviewer for the appreciation of the effort placed in responding to her/his critiques.

REVIEWERS' COMMENTS

Reviewer #1 (Remarks to the Author):

I thank the authors for providing responses and corrections to my specific comments. Based on this, I believe the article is now suitable for publication.

I would recommend to improve the display of Fig. 7D, where dots representing the replicates should be centered on each condition.

Reviewer #2 (Remarks to the Author):

I would like to thank the authors for carefully addressing all comments raised by reviewers. The quality of the manuscript has significantly improved for publication.